# Towards Understanding Adversarial Transferability in Federated Learning

**Yijiang Li**                                                     *yijiangli@ucsd.edu*
*University of California San Diego*

**Ying Gao**                                                      *gaoying@scut.edu.cn*
*South China University of Technology*

**Haohan Wang**                                                 *haohanw@illinois.edu*
*University of Illinois Urbana-Champaign*

**Reviewed on OpenReview:** *https://openreview.net/forum?id=hafnY2PiTn*

## Abstract

We investigate a specific security risk in FL: a group of malicious clients has impacted the model during training by disguising their identities and acting as benign clients but later switching to an adversarial role. They use their data, which was part of the training set, to train a substitute model and conduct transferable adversarial attacks against the federated model. This type of attack is subtle and hard to detect because these clients initially appear to be benign.

The key question we address is: How robust is the FL system to such covert attacks, especially compared to traditional centralized learning systems? We empirically show that the proposed attack imposes a high security risk to current FL systems. By using only 3% of the client's data, we achieve the highest attack rate of over 80%. To further offer a full understanding of the challenges the FL system faces in transferable attacks, we provide a comprehensive analysis over the transfer robustness of FL across a spectrum of configurations. Surprisingly, FL systems show a higher level of robustness than their centralized counterparts, especially when both systems are equally good at handling regular, non-malicious data.

We attribute this increased robustness to two main factors: 1) Decentralized Data Training: Each client trains the model on its own data, reducing the overall impact of any single malicious client. 2) Model Update Averaging: The updates from each client are averaged together, further diluting any malicious alterations. Both practical experiments and theoretical analysis support our conclusions. This research not only sheds light on the resilience of FL systems against hidden attacks but also raises important considerations for their future application and development.

## 1 Introduction

Although Federated Learning (FL) provides a promising solution to collaboratively train models without exchanging data, especially in privacy-concerned areas Li et al. (2022), it is still susceptible to attacks such as data poisoning Huang et al. (2011), model poisoning Bhagoji et al. (2019); Bagdasaryan et al. (2020); Huang et al. (2023), free-riders attack Lin et al. (2019), and reconstruction attacks Geiping et al. (2020); Zhu et al. (2019). It is also vulnerable to adversarial attacks during inference Biggio et al. (2013); Szegedy et al. (2013), including adversarial examples designed to deceive the model Zizzo et al. (2020). Research on robust FL methods against adversarial examples has primarily focused on a white-box setting where attackers have full model access Zhou et al. (2020); Reisizadeh et al. (2020a); Hong et al. (2021); Qiao et al. (2024). However, real-world FL applications, like Gboard Hard et al. (2018), usually restrict such access.

We observe a distinct FL security challenge: **malicious clients may pose as benign contributors, only revealing their adversarial intent post-training**. This setting raises a new security challenge because these clients have access to a subset of the training data, potentially leading to a better surrogate model for transferable attacks. Current FL applications lack effective mechanisms to eliminate such hostile participants Hard et al. (2018), even if selection mechanisms exist, such as Krum Fang et al. (2020); Li et al. (2020a); Bagdasaryan et al. (2020), as attackers do not exhibit hostile behavior during training. After obtaining data, an attacker could train a surrogate model for **transfer-based black-box attacks**.

In this paper, we pioneer an exploration of this practical perspective of FL robustness. Stemming from the above scenarios, we propose a simple yet practical assumption: the attacker possesses a limited amount of the users' data but no knowledge about the target model or the full training set. To assess current FL system robustness and guide future research in this regard, we investigate the *adversarial transferability* under a spectrum of practical FL settings. *Adversarial transferability* refers to the ability of adversarial examples generated from the source model to successfully attack the target model, which measures the amount of threat white-box attack poses to the model and system.

We first evaluate the transferability of adversarial examples generated from different source models to attack a federated-trained model. Then a comprehensive evaluation of practical configurations is conducted to assess the feasibility of our attack. We further investigate two properties of FL: decentralized training and the averaging operation and their correlation with federated robustness. To provide a comprehensive evaluation in a practical aspect, we consider the attack timing, the architecture, and different aggregation methods in our experiments. We have the following findings:

- We find that, while there are indeed security challenges from the novel attack setting, the federated model is more robust under white-box attack compared with its centralized-trained counterpart when their accuracy on clean images is comparable.
- We investigate the transferability of adversarial examples generated from models trained by various numbers of users' data. We observe that without any elaborated techniques such as dataset synthesis Papernot et al. (2017) or attention Wu et al. (2020b), a regularly trained source model with only limited users' data can perform the transfer attack. With ResNet50 on CIFAR10, we achieve a transfer rate of 70% and 80% with only 5% and 7% of users with augmentations and further improve this number to 81% and 85% with AutoAttack Croce & Hein (2020).
- We investigate two intrinsic properties of the FL: the property of distributed training and the averaging operation and discover that both heterogeneity and dispersion degree of the decentralized data as well as the averaging operations can significantly decrease the transfer rate of transfer-based black-box attack.
- To further understand the phenomenon, We provide theoretical analysis to further explain the observations.

## 2 Background

### 2.1 Adversarial Robustness

The adversarial robustness of a model is usually defined as the model's ability to predict consistently in the presence of small changes in the input. Intuitively, if the changes to the image are so tiny that they are imperceptible to humans, these perturbations will not alter the prediction. Formally, given a well trained classifier $f$ and image-label pairs $(x, y)$ on which the model correctly classifies $f(x) = y$, $f$ is defined to be $\epsilon$-robust with respect to distance metric $d(\dot{;})$ if

$$\mathbb{E}_{(x,y)} \min_{x':d(x',x)\leq\epsilon} \alpha(f(x';\theta), y) = \mathbb{E}_{(x,y)}\alpha(f(x;\theta), y) \tag{1}$$

which is usually optimized through maximizing:

$$\mathbb{E}_{(x,y)} \min_{x':d(x',x)\leq\epsilon} \alpha(f(x';\theta), y) \tag{2}$$

where $\alpha$ denotes the function evaluating prediction accuracy. In the case of classification, $\alpha(f(x;\theta), y)$ yields 1 if the prediction $f(x;\theta)$ equals ground-truth label $y$, 0 otherwise. The distance metric $d(\dot{;})$ is usually

approximated by $L_0$, $L_2$ or $L_\infty$ to measure the visual dissimilarity of original image $x$ and the perturbed image $x'$. Despite the change to the input is small, the community have found a class of methods that can easily manipulate model's predictions by introducing visually imperceptible perturbations in images Szegedy et al. (2013); Goodfellow et al. (2014); Moosavi-Dezfooli et al. (2016). From the optimization standpoint, it is achieved by maximizing the loss of the model on the input Madry et al. (2017):

$$\max_\delta \ l(f(x + \delta; \theta), y) \ \text{ s.t. } \ d(x + \delta, x) < \epsilon \tag{3}$$

where $l(\cdot, \cdot)$ denotes the loss function (*e.g.*, cross-entropy loss) for training the model $f$ parameterized by $\theta$. While these attack methods are powerful, they usually require some degrees of knowledge about the target model $f$ (*e.g.*, the gradient). Arguably, for many real-world settings, such knowledge is not available, and we can only expect less availability of such knowledge on FL applications trained and deployed by service providers. On the other hand, the hostile attacker having access to some but limited amount of users' data is a much more realistic scenario. Thus, we propose the following assumption for practical attack in FL: given the data of $n$ malicious users $D_m = \bigcup_{i=1}^n D_i$ where $D_i = \{(x_k, y_k) | k = 1, \cdots, m_i\}^{(i)}$ contains $m_i$ data points, we aim to acquire a transferable perturbation $\delta$ by maximizing the same objective as in Equation 3 but with a surrogate model $f'$ parameterized by $\theta'$ trained by $D_m$:

$$\delta = \arg\max_\delta \ l(f'(x + \delta; \theta'), y) \ \text{ s.t. } \ d(x + \delta, x) < \epsilon \tag{4}$$

We hope to test whether this $\delta$ can be used to deceive the target model $f$ as well.

## 2.2 The Security and Robustness of Federated Learning

**Poisoning and Backdoor Attack.** Poisoning attacks Biggio et al. (2012); Fang et al. (2020) aim to disrupt the global model by injecting malicious data or manipulating local model parameters on compromised devices. In contrast, backdoor attacks Bagdasaryan et al. (2020); Sun et al. (2019); Wang et al. (2023) infuse a malicious task into the existing model without impacting its primary task accuracy Zhang et al. (2023); Huang et al. (2024), including model replacement Chen et al. (2017), label-flipping Fung et al. (2018), fixed-trigger backdoor Dai & Li (2023); Liu et al. (2024) and trigger-optimization Huang (2020); Li et al. (2023a); Nguyen et al. (2024). Defenses against these attacks often involve anomaly detection methods, such as Byzantine-tolerant aggregation Shejwalkar & Houmansadr (2021) (e.g., Krum, MultiKrum Blanchard et al. (2017), Bulyan Guerraoui et al. (2018), Trimmed-mean and Median Yin et al. (2018)), focusing on the geometric distance between hostile and benign gradients. More advanced defenses take detection, aggregation, detection, and differential privacy into consideration Huang et al. (2023); Nguyen et al. (2022). The robustness and attack performance of backdoor attacks is significantly influenced by FL data heterogeneity Zawad et al. (2021).

**Transfer Attack.** Transfer-based adversarial attacks employ the full training set of the target model to train a surrogate model Zhou et al. (2018), a challenging condition to meet in practice, especially in FL where data privacy is paramount. Another line of inquiry delves into the mechanisms of black-box attacks, which exploit the high transferability of adversarial examples even between different model architectures Szegedy et al. (2013); Goodfellow et al. (2014). This transferability is partially attributed to the similarity between source and target models Goodfellow et al. (2014); Liu et al. (2016); Li et al. (2023b), as adversarial perturbations align closely with a model's weight vectors, and different models learn similar decision boundaries for the same task. Tramèr et al. (2017) found that adversarial examples span a large, contiguous subspace, facilitating transferability. Meanwhile, Ilyas et al. (2019) posits that adversarial perturbations are non-robust features captured by the model, and Waseda et al. (2023) utilizes this theory to explain differing mistakes in transfer attacks. Additionally, Demontis et al. (2019); Zhang et al. (2024) reveals that similar gradients in source and target models and lower variance in loss landscapes increase transfer attack probability. Despite transfer attacks being more realistic and practical, not much attention has been focused on this aspect of the safety and robustness of FL. We take the initiative to investigate this area and underscore our setting's distinctiveness and importance compared to others.

**Key Difference 1:** Different from query-based or transfer-based black-box attack, we assume the malicious clients possess the data themselves, impacting the target model during training and attack during inference.

We also present a comparison of our attack setting and the query-based attack in Section 4.3. Note that our attack setting doesn't contradict the query-based attack. In fact, we can perform with both if the FL system allows a certain number of queries, which we leave to future works to explore.

**Key Difference 2:** Poisoning attack or backdoor attack manipulates the parameters update during target model training which can be defended by anomaly detection. Moreover, in practice, despite clients preserving the training data locally, the training procedure and communication with the server are highly encapsulated and encrypted with secret keys, which is even more unrealistic and laborsome to manipulate. Our attack setting circumvents this risk since no hostile action is performed during the training but successfully boost the attack possibility during inference time.

**Significance:** Besides the potential data leakage by malicious participants, we also emphasize that despite, ideally, each participant having access to the global model, in real-world applications (e.g. Gboard Hard et al. (2018)), the infrastructure provider will impose additional protection such as encryption or encapsulation over the local training. For instance, Google's Gboard provides next-word prediction with FL, which requires users to install an app to participate. For an adversary, it's impractical to obtain the global model without breaking or hacking the app or hijacking and decrypting the communication, despite all the things happening "locally". We believe this is much more difficult and laborsome than our setting which significantly boosts the transferability by simply acting as a benign.

## 3 Investigation Setup and Research Goals

**GOAL 1:** We aim to investigate the possibility of a transfer attack with limited data and validate whether it is possible and practical for the attacker to lay benign during the training process and leverage the obtained data to perform the adversarial attack.

**GOAL 2:** We aim to explore how different degrees of decentralization, the heterogeneity of data and the aggregation, *i.e.* average affect the transferability of the adversarial examples against the FL model in a practical configuration.

### 3.1 Experiment Setup

**Threat Model.** Following (Zizzo et al., 2020), we use PGD (Madry et al., 2017) with 10 iterations, no random restart, and an epsilon of 8 / 255 over $L_\infty$ norm on CIFAR10. For experiments on ImageNet200, we use PGD (Madry et al., 2017) of the same setup but with an epsilon of 2 / 255.

**Settings.** We first build up the basic FL setting. We split the datatset into 100 partitions of equal size in an iid fashion. We adopt two models for the experiments: CNN from (McMahan et al., 2017) since it is commonly used in the FL literature and the widely used ResNet50 (He et al., 2016) which represents a more realistic evaluation. We conduct training in three paradigms: the centralized model, the federated model and the source model with a limited number of clients' data. For the federated model, we use SGD without momentum, weight decay of 1e-3, learning rate of 0.1, and local batch size of 50 following (Acar et al., 2021). We follow the cross-device setting and use a subset of clients in each round of training. We use a 10% as default where not specified. We train locally 5 epochs on ResNet50 and 1 epoch on CNN. For centralized and source model training, we leverage SGD with a momentum of 0.9, weight decay of 1e-3, a learning rate of 0.01 and batch size of 64. For adversarial training, we use the same setting as centralized and leverage PGD to perform the adversarial training. We refer to (Zizzo et al., 2020) for the details of adversarial training. All experiments are conducted with one RTX 2080Ti GPU.

**Metrics.** We report Accuracy (Acc) and adversarial accuracy (Adv.Acc) for the performance and the robustness of white-box attack, and for adversarial transferability, we report transfer accuracy (T.Acc) and transfer success rate (T.Rate) as detailed in 3.2.

### 3.2 Adversarial Transferability in Federated Learning

To define the transferability of adversarial examples, we first introduce the definition of the source model, target model and adversarial example. The source model is the surrogate model used to generate adversarial examples while the target model is the target aimed to attack. Given the validation set $x = \{(x_i, y_i)\}$, source model $f'$, target model $f$ and adversarial perturbation function $adv(\cdot, \cdot)$ (*e.g.*, PGD), we first define the following sets:

$$s1 = \{x_i | f'(x_i) = y_i\},$$
$$s2 = \{x_i | f'(adv(x_i, f')) \neq y_i\},$$
$$s3 = \{x_i | f(x_i) = y_i\},$$
$$s4 = \{x_i | f(adv(x_i, f')) \neq y_i\}$$

Adversarial examples are defined as those samples that are originally correctly classified by model $f'$ but are misclassified when the adversarial perturbation is added, i.e., $s1 \cap s2$. Adversarial transferability against the target model refers to the ability of adversarial examples generated from the source model to attack the target model (become an adversarial example of the target model). We define transfer rate (T.Rate) and transfer accuracy (T.Acc) to measure the adversarial transferability: $\text{T.Rate} = \frac{||s1 \cap s2 \cap s3 \cap s4||}{||s1 \cap s2 \cap s3||}$, $\text{T.Acc} = 1 - \frac{||s4||}{||x||}$ where $|| \cdot ||$ denotes the cardinality of a set. We are the first one to propose and use the Transfer Rate metric to measure the transferability of adversarial examples which measures the transferability of the surrogate model by measuring the portion of transferable examples. This serves as a complimentary to accuracy as plain accuracy fails to accurately measure robustness.

## 4 Experiments

### 4.1 Robustness with Comparable Accuracy

In order to provide a preliminary understanding about the robustness of the FL model, we train the centralized model for 200 epochs and the federated model for 400 rounds resulting in a decent accuracy of over 90% (see the regular column of Tab. 1). For the CNN model, we train 200 epochs for the centralized model and 600 rounds for the federated model to achieve an accuracy of over 75%.

We can observe that the federated model's clean and adversarial accuracy is lower than its centralized counterpart, aligned as the result in (Zizzo et al., 2020). However, we conjecture that such an increase in adversarial accuracy is not attributed to the intrinsic robustness of the centralized model but largely due to its high clean accuracy. To validate this hypothesis and facilitate a fair comparison between the two paradigms, we early-stop both models when their clean accuracy reaches 90% (75% for CNN) and report the results in the same-acc column of Tab. 1. We early stop at 80% for adversarial training (72% for CNN). We can see that when both models reach a comparable clean accuracy, the FL model shows greater robustness against white-box attacks compared with the centralized model.

Table 1: Centralized and federated model under white-box attack. We can see that, with comparable clean accuracy, the FL model shows greater robustness against white-box attacks compared with the centralized model. This observation is consistent across different datasets and model architectures.

| Paradigm | Architecture | same-acc | | regular | |
|---|---|---|---|---|---|
| | | Acc | Adv.A | Acc | Adv.A |
| | CIFAR10 | | | | |
| centralized | R50 | 90.20 | 0.01 | 95.24 | 0.40 |
| | R50 (adv) | 81.23 | 23.27 | 89.46 | 46.09 |
| | CNN | 75.06 | 1.24 | 82.41 | 0.35 |
| | CNN (adv) | 73.15 | 20.89 | 76.78 | 28.92 |
| federated | R50 | 90.29 | 0.05 | 92.31 | 0.02 |
| | R50 (adv) | 80.05 | 36.44 | 81.05 | 39.11 |
| | CNN | 75.09 | 3.68 | 76.83 | 3.98 |
| | CNN (adv) | 72.85 | 25.5 | 72.87 | 24.35 |
| | ImageNet200 | | | | |
| centralized | R50 | 55.05 | 3.68 | 65.79 | 8.41 |
| | R50 (adv) | 50.04 | 31.20 | 55.59 | 32.84 |
| federated | R50 | 55.03 | 13.42 | 60.59 | 15.68 |
| | R50 (adv) | 50.18 | 38.31 | 54.92 | 41.13 |

To further validate our hypothesis, we perform the experiment on a much larger and more realistic dataset, *i.e.* ImageNet200. We early stopped both models at 55% accuracy. Results can be seen in the bottom two rows of Tab. 1. We can see that FL models demonstrate superior robustness against white-box attacks compared with the centralized model on both same-acc and regular settings.

## 4.2 Robustness Against Transfer Attack

We turn to the black-box attack which is more practical and realistic in real-world applications. We explore the examples generated by two different training paradigms and their transferability to different models. Since the similarity of decision boundary and clean accuracy influences and reflects the transferability between models (Goodfellow et al., 2014; Liu et al., 2016; Demontis et al., 2019), we early stop both federated and centralized models. For CIFAR10, we early stop both models at 90% of accuracy (75% for CNN). For ImageNet200, we follow Section 4.1 and early stop at 55%. We follow this training setting for the rest of this paper.

Table 2: T.Rate and transfer accuracy of PGD attack between pairs of models using various training paradigms. The row and column denote the source and target model respectively. For each cell, the left is the transfer accuracy and the right is the T.Rate.

|  |  | federated | centralized |
|---|---|---|---|
| | | CIFAR10 | |
| R50 | federated | 0.15 / 99.83 | 2.01 / 97.67 |
| | centralized | 24.28 / 71.94 | 7.41 / 91.48 |
| CNN | federated | 19.31 / 76.32 | 21.59 / 71.84 |
| | centralized | 30.57 / 56.59 | 22.62 / 68.19 |
| | | ImageNet200 | |
| R50 | federated | 2.29 / 95.60 | 6.52 / 86.74 |
| | centralized | 22.72 / 54.00 | 8.27 / 83.13 |

Tab. 2 shows that the adversarial examples generated by the federated model are **highly transferable** to both the federated and centralized model while adversarial examples generated by the centralized model exhibit less transferability. The T.Rate of federated-to-centralized attack is even larger than centralized-to-centralized attack. Secondly, T.Rate of adversarial examples between models trained under same paradigms is larger than models trained under different paradigms, which can be attributed to the difference of the two training paradigms, *e.g.*, the discrepancy in the decision boundary (Goodfellow et al., 2014; Liu et al., 2016) or different sub-space (Tramèr et al., 2017).

## 4.3 Transfer Attack with limited data

In this section, we comprehensively evaluate the practicality and plausibility of our proposed attack setting, *i.e.* transfer attack with limited data. We present the overview of our attack setup in Algorithm 1. To simulate this scenario, we fix the generated partition used in the federated training and randomly select a specified number of users as malicious clients whose data is available for performing the attack. To perform the transfer attack, we train a surrogate model in a centralized manner with the collected data. Training details are specified in Section 3.1.

---

**Algorithm 1:** Our Attack as Benign Setting

---

**Input:** A set of $N$ clients $\{C_1, C_2, \ldots, C_N\}$ where $M$ clients $\{C_1, C_2, \ldots, C_M\}$ are corrupted by attacker, datasets $X_i$ for each client $C_i$, sampling rate $K$, total communication round $T$, local iteration number $E$ and surrogate model training iteration number $T_s$.

**Output:** Adversarial perturbation $\delta$ on example $x$

---

/* Normal Federated Training with corrupted clients acting as benign */
initialize FL model $f_0$; **for** *each round* $t = 1, 2, \cdots, T$ **do**
    Server samples $K$ devices $S_t$ and distributes the global model $f_t$
    **for** *each client* $C_i$ *in* $S_t$ **do in parallel**
        $f_{t+1}^i \leftarrow \text{ClientUpdate}(f_t, X_i)$
    **end**
    $f_{t+1} \leftarrow Aggregate(f_{t+1}^1, \cdots, f_{t+1}^K)$
**end**
/* Train surrogate model */
Adversary collects data from corrupted clients $\{C_1, C_2, \ldots, C_M\}$; initialize surrogate model $f_0'$;
**for** *each iter* $t = 1, 2, \cdots, T_s$ **do**
    $f_t' \leftarrow \text{ClientUpdate}(f_{t-1}', \bigcup_{i=1,\cdots,M} X_i)$
**end**
$\delta \leftarrow \text{AdvPerturb}(f', x)$;

---

One of the key differences between the proposed setting and the conventional transfer-based attack is the amount of available data to train the substitute (source) model, which is a key factor for a successful attack since one would reason that more data will lead to a higher success rate. This is measured by the number of

clients used to train the source model. To provide an overview of the transferability of adversarial examples generated by the source model trained with different numbers of clients, we plot their relation in Fig. 1. We have the following observations:

- **Observation 1.** T.Rate increases as the number of users increases, which is consistently observed in both centralized and federated models.
- **Observation 2.** With only 20% of clients the source model achieves an T.Rate of 90% and 50% with ResNet50 and CNN respectively. We notice that with ResNet50, the T.Rate of 20% clients is even larger than a transfer attack with full training data (71.94% T.Rate). With CNN, the source model trained with 20% clients can achieve 50% T.Rate which only lags behind the transfer attack by 6% (56.59%).

From observation 2, we can see that the proposed attack can achieve comparable or even better T.Rate or lower T.Acc. **Consequently, we can conclude that the proposed attack setting with limited data is likely to cause significant security breaches in the current and future FL applications.** To further explain observation 1 and an intriguing phenomenon that the T.Rate of ResNet50 model rises to the peak and then decreases, we provide the following hypothesis: When the number of clients used to train the source model is small, the clean accuracy of the source model is also low, leading to a large discrepancy in the decision boundary. Increasing the number of users used in the source model minimizes such discrepancy until the amount of data is sufficient to train a source model with similar accuracy. At this point, the difference between the federated and centralized (Caldarola et al., 2022) becomes the dominant factor affecting the transferability since the source model is trained in the centralized paradigm.

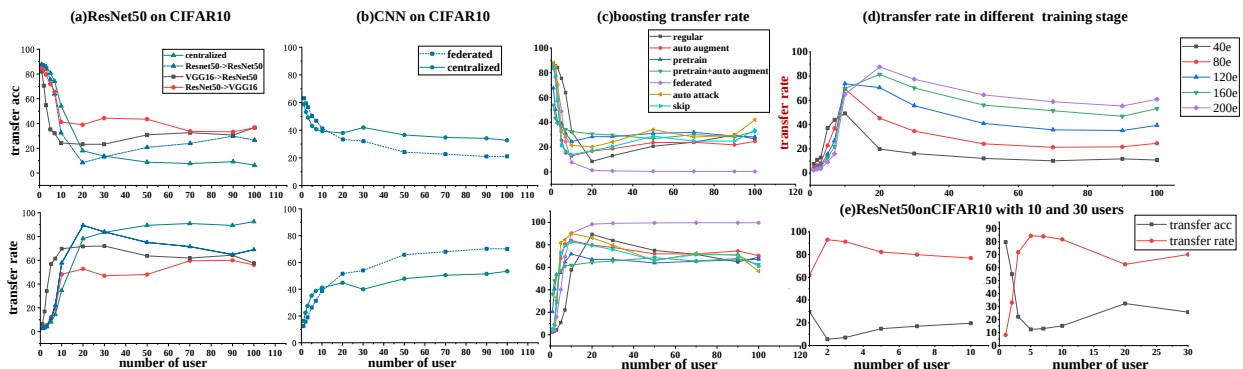

Figure 1: Attack with data from a limited number of users. (a) We show the transfer rate of our attack with ResNet50 on the CIFAR10 dataset. We additionally experiment with attackers of different architectures. (b) experiment with CNN on CIFAR10 dataset. (c) we boost the performance of our attack with standard augmentation and pretraining techniques. (d) we perform our attack on different training stages. (e) we provide experiments on easier scenarios with 10 and 30 users, which further demonstrate the threat our attack poses.

**Training surrogate model in a federated manner.** To validate the conjecture above, we train the surrogate model in a federated manner. Specifically, since we know each sample's client, we partition the collected data from malicious clients as the federated model and train the source model in a federated manner. Results can be seen in (c) of Fig. 1: with a federated source model, the T.Rate can be slightly boosted at the beginning (limited number of clients) but continue to increase as the number of users increases and finally contributes to a significantly high T.Rate of 99%. This demonstrates our conjecture and also shows that if the hostile party trains the surrogate model in a federated manner, the T.Rate can be further increased. We further evaluate whether the partition information is crucial to the higher transferability in Appendix A.

**Boosting transfer rate.** Following the conjectures above, we propose several effective approaches to enhance the transferability of the surrogate model trained by the malicious data only. We first boost the T.Rate of the source model trained with limited data with data augmentation and model pretraining. Without loss of generality, we leverage AutoAugment (Cubuk et al., 2018) and ImageNet pretrain. We believe other forms of augmentation and pretrained weights will exhibit similar effects. From (c) of Fig. 1, we can see, with these techniques we can successfully increase the T.Rate of 1% and 2% of clients from around 3% to 36% and 48% respectively. With 7% or 10% of clients' data, the proposed attack setting achieves a high T.Rate of more

than 80% (10% higher than the transfer-based attack). With simple training techniques, malicious clients can attack with more than 40% success rate with one or more clients and 80% with 7 to 10 clients.

**Advanced Transfer Attack.** To further show the threat posed by our attack, we leverage more advanced attacks to replace the default PGD attack as shown in (c) of Fig. 1: both AutoAttack (Croce & Hein, 2020) and skip attack (Wu et al., 2020a) achieve a transfer rate of over 80% with only 3 and 10 users respectively. Noticeably, AutoAttack achieves the highest transfer rate of 85% with 10% of the users.

**Comparison with query-based attack** As elaborated in Sec. 2, our transfer attack with limited data is similar but different from the query-based black-box attack, as we assume the malicious clients possess a portion of the original training data. Query-based black-box attack, on the other hand, aims to attack the target model with limited queries to the target model. Through these queries, the source model manages to optimize the decision boundary towards the target model. However, most APIs and FL-based systems require charges to access the service or equip anomaly detection that detects multiple or malicious queries.

We provide a comparison of our proposed attack setting and the query-based black-box attack. To facilitate a fair comparison, we follow the same experiment setting to train the query-based model. We plot the comparison in (a) of Fig. 2. Despite query-based outperforming the default of our proposed attack by a slight margin at the cost of expensive queries, our attack combined with federated training outperforms the query-based attack consistently at all configurations. It is also pertinent to note that facilitating the queries for data from only one client necessitates the execution of 500 requests to the target model (in a 100-client partition). Further, the query cost demonstrates a proportional escalation as the number of required queries increases, which may be regarded as impracticable in tangible operational contexts. Moreover, we underscore that our attack does not counteract the query-based attack. In fact, we can perform both if the FL system allows a certain number of queries. We believe that, combined with a limited number of queries, our attack strategy will be augmented to attain a higher transfer rate since the queries from the target model can help the surrogate model learn a decision boundary that more congruently aligns with that of the target model.

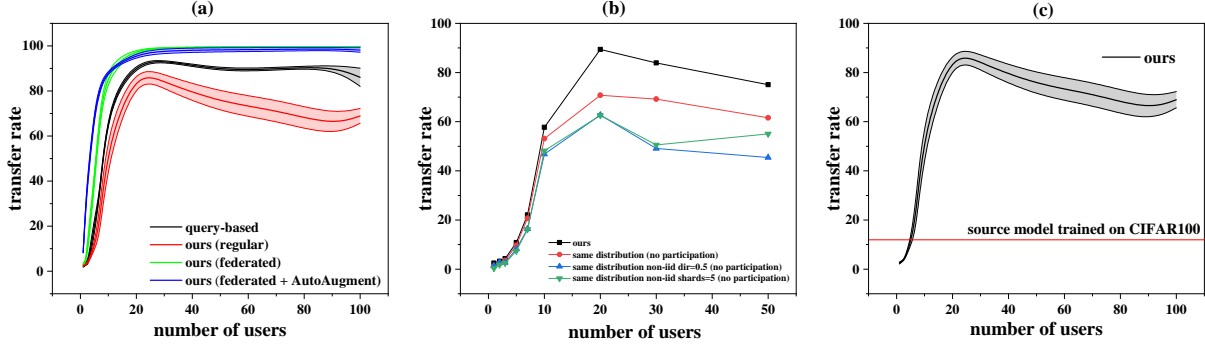

Figure 2: (a) Comparison with query-based black-box attack. (b) Attack with the surrogate model trained with the same distribution (no participation in the FL training) (c) Attack with the surrogate model trained with similar knowledge, i.e. CIFAR100.

**Practical evaluation.** To practically evaluate our attack setting, we consider two cases: 1. the malicious clients can only participate in some periods during the FL training and 2. the attacker has no knowledge of the model deployed (i.e. unknown architecture). We simulate these situations by attacking in different training stages and with different architectures, as shown in (d) of Fig. 1.

**Is Data Distribution or Similar Knowledge Sufficient?** We emphasize the significance of the process that malicious clients stay benign during training and affect the training with their own data. We demonstrate in this section that this obtained training data is crucial to a successful attack.

We first show that using a similar distribution is not as good as the actual training data to train a surrogate model, as shown in (b) of Fig. 2. We note that the inherent heterogeneity of FL training will further impose challenges. The distinct distribution of each client's data makes it nearly impossible to approximate an overall training distribution due to the scale and variability of the data.

Training the surrogate model with a dataset of similar knowledge or characteristics is also ineffective. We simulate this by using the CIFAR100 dataset to train a surrogate model and perform the attack against the FL model trained on the CIFAR10 dataset, as shown in (c) of Fig. 2. We also want to emphasize that our approach to obtaining the training data by acting benignly during training is practically impossible to defend against as there is no way to distinguish a benign client and a malicious client if they stay benign.

**More Configurations.** We also conduct experiments with 10 and 30 users as shown in (e) of Fig. 1 which is a relatively simpler setting. Specifically, we emphasize that when the federated model is trained on 10 and 30 uses, our attack achieves the highest 90% transfer rate with only 2 and 5 malicious clients respectively. When there are only 1 malicious client in the 10-user setting and 3 malicious clients in the 30-user setting, we can achieve over 60% transfer rate and over 70% transfer rate. We further perform experiments on CIFAR100, SVHN and a much larger and more realistic dataset ImageNet200, as demonstrated in (a) of Fig. 5 where similar trends are demonstrated. These results emphasize the threat of our attack setting.

# 5 Two intrinsic properties contributing to transfer robustness

To fully understand how adversarial examples transfer between centralized and federated models, we study two intrinsic properties of FL and its relation with transfer robustness. To protect the privacy of clients and leverage the massive data from user-end, FL utilizes distributed data to train a global model through local updates and aggregation at the server (McMahan et al., 2017). As a consequence, the heterogeneity of the distributed data and the aggregation operation is the core component of an FL method. In this section, we study how these two properties affect the transfer robustness of the FL model.

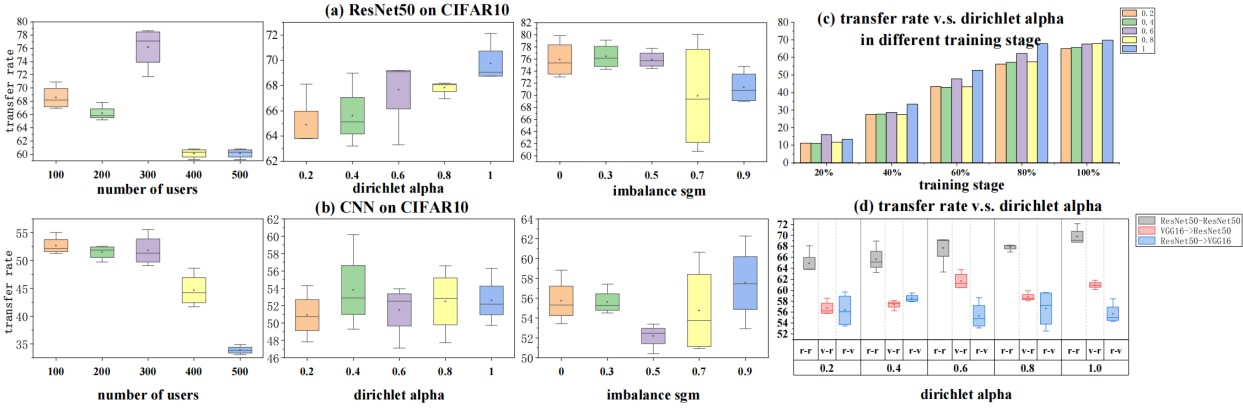

Figure 3: T.Rate vs. data of different heterogeneity and dispersion degree. (a): top 3 are results of ResNet50; (b): bottom 3 are results of CNN; Left: T.Rate as a function of the number of users in federated training; Middle: T.Rate as a function of dirichlet alpha; Right:T.Rate as a function of unbalanced sgm.

## 5.1 Decentralized training and data Heterogeneity

This section aims to explore the relationship between the degree of heterogeneity and transfer robustness.

**Control the degree of dispersion and heterogeneity.** To explore the impact of distributed data on adversarial transferability, we control decentralization and heterogeneity through four indexes. By varying the number of clients in the partition, we alter the degree of dispersion of distributed data. We provide two approaches to control the heterogeneity of the distributed data. 1. Change the number of maximum classes per client (McMahan et al., 2017). 2. Change the alpha value of Dirichlet distributions ($\alpha$) (Wang et al., 2020; Yurochkin et al., 2019) (smaller $\alpha$ means a more non-iid partition) and the log-normal variance (sgm) of the Log-Normal distribution (larger variance denotes more unbalanced partitions) used in unbalanced partition (Zeng et al., 2021). We leverage the FedLab framework to generate the different partitions (Zeng et al., 2021). For simplicity, we leverage the centralized trained model as the source model for the rest of the experiments.

**Degree of decentralization reduces transferability.** We first explore the relation between decentralization and transferability. To control the degree of decentralization, we generate partitions with different numbers of clients and train target federated models on these partitions. Then we perform the transfer attack using the centralized model as the source model. As seen in Fig. 3 (left of (a) and (b)), we can observe that despite some fluctuations, T.Rate drops with an increasing number of users, which demonstrates that more decentralized data leads to lower transferability. We provide statistical testing for the correlation coefficient in Appendix C to further validate the above observation. With the Spearman correlation coefficient, we report a significant negative correlation on both ResNet50 and CNN between the degree of decentralization and T.Rate under a significance level of 0.1 with $p$-value=0.03 and 0.01 respectively for ResNet50 and CNN. We also visualize linear regression to fit the negative correlation between the degree of decentralization and T.Rate the in Appendix H.

**Data heterogeneity affects transfer attack.** As discussed in Sec. 5.1, we provide two approaches for controlling the heterogeneity of the data. First, we alter the alpha values of the Dirichlet distributions to generate heterogeneous data of different degrees. As per the middle plot of (a) and (b) in Fig. 3, we can see that T.Rate increases as $\alpha$ increases.

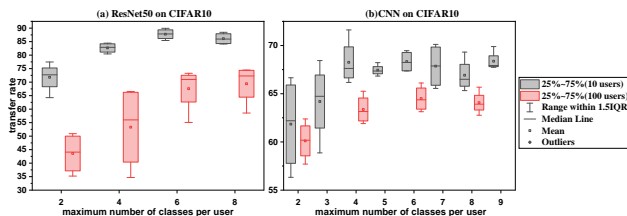

We also alter the maximum number of classes per client to generate heterogeneous data of different degrees. Results are reported in Fig. 4. We can observe from Fig. 4 a clear increase trend in both the 10-user partition and the 100-user partition setting with ResNet50 and CNN. As the degree of heterogeneity decreases, the transferability increases. Experiments in both settings can illustrate our findings. This proves that our observations hold to wider circumstances and scenarios.

Figure 4: Transfer rate v.s. maximum number of classes per client; (a): Results of ResNet50 on CIFAR10 dataset; (b): Results of CNN on CIFAR10 dataset; It is noteworthy to observe that the transfer rate in the 10-user setting persistently surpasses that in the 100-user setting. This further substantiates the proposition that more decentralized training leads to lower adversarial transferability for the federated model.

We further explore whether unbalanced data (i.e. different clients possess different numbers of samples) leads to a decrease in transferability in Fig. 3 (right of (a) and (b)). We can observe that a larger variance leads to a lower transfer rate, meaning that unbalanced data also contribute to higher robustness.

To validate the above observation, we provide statistical testing for the correlation coefficient in Appendix C. With the Spearman correlation coefficient, we report a significant negative correlation on ResNet50 between log-normal variance and T.Rate under a significance level of 0.1 ($p$-value=0.05). We report a significant correlation on all results with a level of 0.05 except the CNN experiments with different Dirichlet $\alpha$ and unbalanced sgm. Visualization of linear fitting is in Appendix H. To provide a more practical evaluation, we follow the setting in practical evaluation of Sec. 4.3 and evaluate the above findings in different training stages and with different architectures. As per (c) and (d) of Fig. 3, we can observe a similar correlation and trends between Dirichlet alpha and transfer rate. This further illustrates that our findings hold in various settings and configurations.

## 5.2 Averaging Leads to Robustness

We explore the other core property, averaging operation, of FL and its correlation with transfer robustness. To change the degree of averaging in FL, we alter the number of clients selected to average at each round. To comprehensively illustrate this finding, we use two source models to attack. The first one is a centralized model trained with all the data. The Second surrogate model is trained with 30% of all the clients' data, simulating the attack in Sec. 4.3. We plot the relation of T.Rate and #averaged users in (a) to (d) of Fig. 6. Both CNN and ResNet50 exhibit a decreasing trend as more users participate in the averaging operation. This demonstrates that the averaging operation contributes to the robustness of the FL model and more clients to average per round leads to higher transfer robustness. We provide statistical testing to validate

the correlation in Appendix C. With the Spearman correlation coefficient, we report a significant negative correlation in all four experiments (all p-values are less than .001).

To provide a more practical evaluation, we follow the setting in practical evaluation of Sec. 4.3 and evaluate the above findings in different training stages and with different architectures. As shown in (e), (f) of Fig. 6, we can observe a similar trend as mentioned above. This demonstrates that this phenomena hold to more practical settings and wider configurations. We further illustrate that this observation generalizes to different aggregation methods in Appendix B.

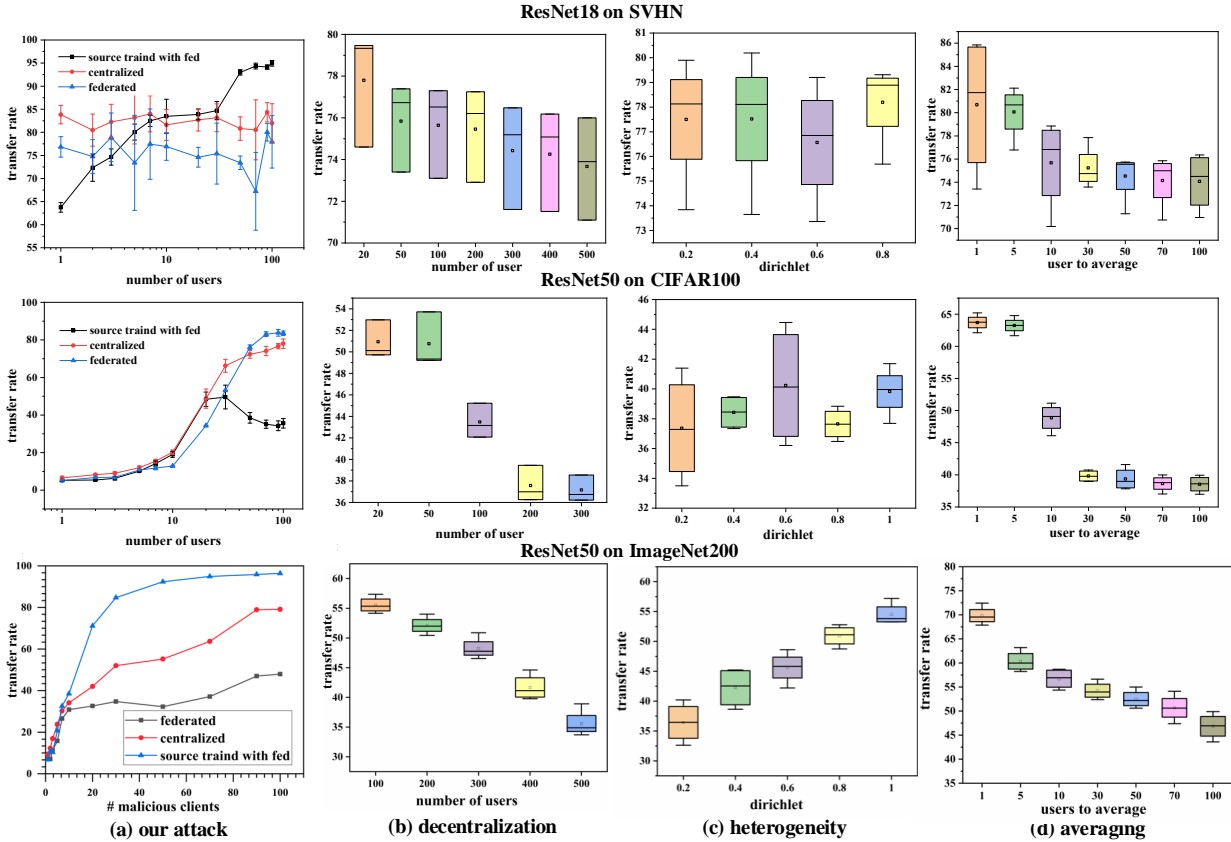

Figure 5: Additional experiments on SVHN, CIFAR100 and ImageNet200. (a) Our attack with ResNet18 on SVHN (first row), ResNet50 on CIFAR100 (second row) and ResNet50 on ImageNet200 (third row). (b) How decentralization relates to the transfer robustness of FL model on SVHN (first row), CIFAR100 (second row) and ImageNet200 (third row) datasets. (c) How heterogeneity relates to the transfer robustness of FL model on SVHN (first row), CIFAR100 (second row) and ImageNet200 (third row) datasets. (d) How averaging operation relates to the transfer robustness of FL model on SVHN dataset (first row), CIFAR100 (second row) and ImageNet200 (third row) datasets.

## 5.3 Discussion

We summarize the above investigations as the below take-home messages and provide implications for understanding adversarial transferability in FL and secure FL applications:

- The heterogeneous data and a large degree of decentralization both result in lower transferability of adversarial examples from the surrogate model. → The attacker can benefit from closing the discrepancy between the surrogate model and the target model (*e.g.*, train the surrogate model in a federated manner).
- With more clients to average at each round, the federated model becomes increasingly robust to black-box attacks. → Defenders can benefit from increasing the number of clients selected at each round to average.
- In addition, we also identify a different, simpler, but practical attack evaluation for FL, which can serve as the standard robustness evaluation for future FL applications.

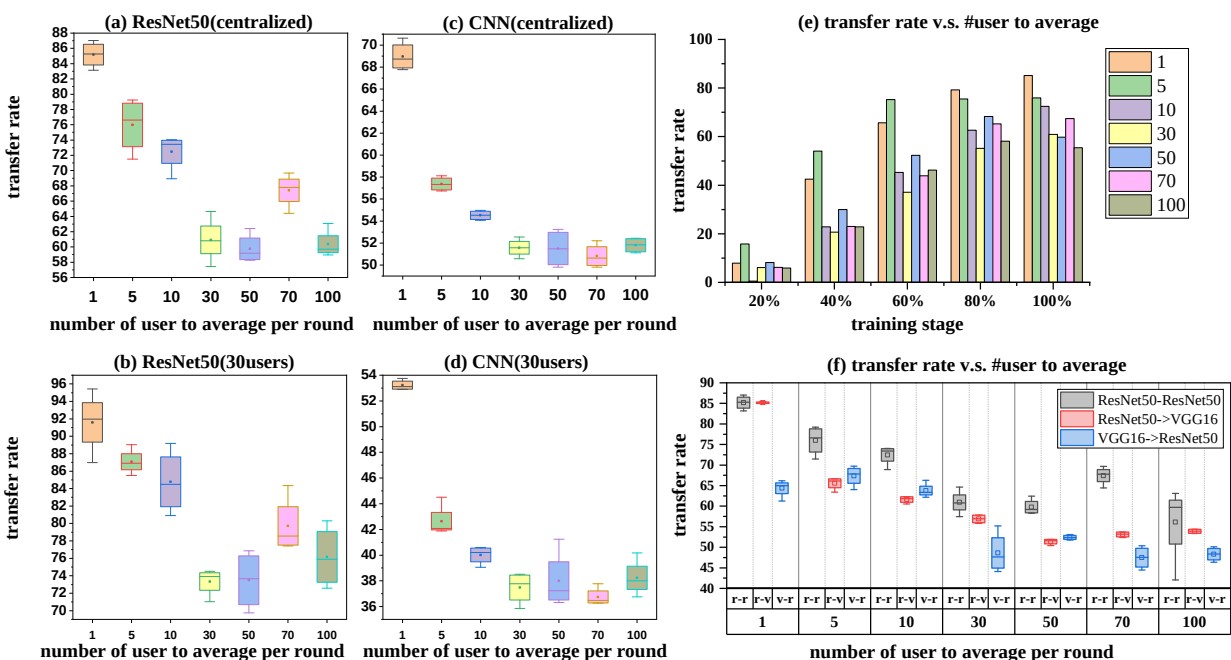

Figure 6: Transfer rate v.s. different number of clients selected to average in each round; (a) ResNet50 results with source model trained in centralized manner with full data; (b) ResNet50 results with source model trained with 30 users; (c) CNN results with source model trained in centralized manner with full data; (d) CNN results with source model trained trained with 30 users;

## 6 Supporting Theoretical Evidence

**Notation.** We use $(X, Y)$ to denote a dataset, where $X \in \mathbb{R}^{n \times p}$ and $Y \in \mathbb{R}^n$. We use $(x, y)$ to denote a sample following Section 2. We consider the problem of federated learning optimization model: $\min_\theta \{l(\theta) \triangleq \sum_{k=1}^{N} p_k l_k(\theta)\}$, where $l_k(\theta) = \frac{1}{n_k} \sum_{j=1}^{n_k} l(f(x; \theta), y)$, $N$ is the total number devices and $n_k$ is the number of samples possessed by device $k$. $p_k$ is the weight of device $k$ with the constraint that $\sum p_k = 1$. Following Li et al. (2020b). we assume the algorithm performs a total of $T$ stochastic gradient descent (SGD) iterations and each round of local training possesses $E$ iterations. $\frac{T}{E}$ is thus the total communication times. At each communication, a maximal number of $K$ devices will participate in the process. We quantify the degree of non-iid using $\Gamma = l_\star - \sum_{k=1}^{N} p_k l_{k\star}$ where $l_\star$ and $l_{k\star}$ is the minimum value of $l(\theta)$ and $l_k(\theta)$ respectively. We use the relative increase of adversarial loss Demontis et al. (2018) to measure the transferability for easier derivation, which is defined as the loss of the target model of an adversarial example, simplified through a linear approximation of the loss function: $T = l(f(x + \hat{\delta}; \theta), y) \approx l(f(x; \theta), y) + \hat{\delta}^T \nabla_x l(f(x; \theta), y)$, where $\hat{\delta}$ is some perturbation generated by the surrogate model, which corresponds to the maximization of an inner product over an $\epsilon$-sized ball under the above linear approximation:

$$\hat{\delta} \in \arg\max_{\|\delta\|_p < \epsilon} l(f'(x + \delta; \theta'), y), \quad \max_{\|\delta\|_p < \epsilon} \delta^T \nabla_x l(f'(x; \theta'), y) = \epsilon \|\nabla_x l(f'(x; \theta'), y)\|_q$$

where $\|\|_q$ and $\|\|_p$ are dual norm.

Without loss of generality, we take $p = 2$ and gives optimal $\hat{\delta} = \epsilon \nabla_x l(f'(x; \theta'), y) / \|\nabla_x l(f'(x; \theta'), y)\|_2$ from the surrogate model. Substituting it back to the Equation 6 we have the loss increment, bounded by the loss of the white-box attack:

$$\Delta l = \epsilon \frac{\nabla_x l(f'(x; \theta')^T \nabla_x l(f(x; \theta), y)}{\|\nabla_x l(f'(x; \theta'), y)^T\|_2} \leq \epsilon \|\nabla_x l(f(x; \theta), y)\|_2$$

We define the relative increase in loss in the black-box case compared to the white box as $R(x, y)$, which we show has a lower bound.

$$R(x, y, \theta, \theta') = \frac{\nabla_x l(f'(x; \theta'), y)^T \nabla_x l(f(x; \theta), y)}{\|\nabla_x l(f'(x; \theta'), y)\|_2 \|\nabla_x l(f(x; \theta), y)\|_2} \tag{5}$$

Then we provide a low bound for $R$.

**Theorem 6.1.** *With Assumptions 2-8, we have:*

$$\mathbb{E}[R(x, y, \theta_\star, \theta')] \geq \frac{2\mu(\gamma + T - 1)\theta_\star^T \theta_\star}{4(B + C)\kappa + \mu^2 \gamma \kappa \mathbb{E}\|\theta_1 - \theta_\star\|^2} \tag{6}$$

*where $L, \mu, \sigma_k, G$ are defined in the assumptions, $\kappa = \frac{L}{\mu}$, $\gamma = \max\{8\kappa, E\}$, $B = \sum_{k=1}^N p_k^2 \sigma_k^2 + 6L\Gamma + 8(E-1)^2 G^2$ and $C = \frac{4}{K} E^2 G^2$. $\theta_1$ is the parameter after one step update of SGD. $\theta_\star$ is the optimal parameter for the centralized model.*

We refer to Appendix D for proof.

**Corollary 6.2.** *We derive the lower bound of the expectation of $R(x, y, \theta_\star, \theta'')$ by setting $E = 1$ and $K = 1$, where $\theta''$ represents the centralized source model.*

$$\mathbb{E}[R(x, y, \theta_\star, \theta')] \geq \frac{2\mu(\gamma + T - 1)\theta_\star^T \theta_\star}{4(\sum_{k=1}^N p_k^2 \sigma_k^2 + 6L\Gamma + 4G^2)\kappa + \mu^2 \gamma \kappa \mathbb{E}\|\theta_1 - \theta_\star\|^2}$$

**Remark 6.3.** The difference between the lower bound of FL (Lemma D.1) and the centralized model (Corollary 6.2) lies in the denominator. With FL model,

$$B + C = \sum_{k=1}^N p_k^2 \sigma_k^2 + 6L\Gamma + 8(E-1)^2 G^2 + \frac{4}{K} E^2 G^2,$$

while centralized gives a smaller

$$B + C = \sum_{k=1}^N p_k^2 \sigma_k^2 + 6L\Gamma + 4G^2,$$

leading to a larger lower bound. Thus, the transferability of adversarial examples generated by the surrogate centralized model to attack the federated model is less than that when attacking a centralized model.

Remark 6.3 supports the empirical findings in Section 4.2.

**Remark 6.4.** With Theorem 6.1, we can see the degree of non-iid $\Gamma$ lies in the denominator of the lower bound, meaning that the larger the degree of non-iid among devices, the less the transferability of examples generated by centralized surrogate model.

Remark 6.4 aligns with our empirical findings in Section 5.1, which will provide more insights to future research on federated adversarial robustness.

**Theorem 6.5.** *Let $\mathcal{T}$ denote the train set $\mathcal{T} = (X, Y)$ sampled from some data distribution and $x'$ denote the adversarial example following some adversarial distribution. Let $f(\cdot, \mathcal{T})$ be the model trained with dataset $\mathcal{T}$ and $\bar{f}(\cdot) = \mathbb{E}_{\mathcal{T}}[f(\cdot; \mathcal{T})]$ is the expectation of the model trained on dataset $\mathcal{T}$. We denote the an average of $n$ models as $f_n = \frac{1}{n} \sum_{i=1}^n f_i(\cdot; \mathcal{T}_i)$. Then we have:*

$$\mathbb{E}_{x', y} \mathbb{E}_{\mathcal{T}}[\|y - f_n(x')\|_2^2] = \mathbb{E}_{x', y}[\|y - \bar{f}(x')\|_2^2] + \frac{1}{n} \mathbb{E}_{x', \mathcal{T}}[\|\bar{f}(x') - f(x', \mathcal{T})\|_2^2]$$

**Remark 6.6.** In Theorem 6.5, $\mathbb{E}_{x', y}[\|y - \bar{f}(x')\|_2^2]$ and $\mathbb{E}_{x', \mathcal{T}}[\|\bar{f}(x') - f(x', \mathcal{T})\|_2^2]$ only depends on $f$ and are fixed with respect to $n$. Thus, as $n$ becomes larger, the expected error decreases.

Remark 6.6 supports the observation in Section Section 5.2.

# 7 Conclusion

We explore the potential for malicious clients to masquerade as benign entities in Federated Learning, then exploit this position to launch transferable attacks. We provide a thorough investigation of the proposed

attack with limited data setting. Our evaluation shows that limited data can yield a comparable transfer rate to a full-dataset attack.

To fully understand how adversarial examples transfer between centralized and federated models, we further study two intrinsic properties of FL and its relation with transfer robustness. We discover that decentralized training, heterogeneous data, and averaging operations enhance transfer robustness and reduce the transferability of adversarial examples. We provide evidence from both the perspective of empirical experiments and theoretical analysis.

Our findings have implications for understanding the robustness of federated learning systems and poses a practical question for federated learning applications.

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

# A  Attack with Same or Different Partition

We show in Section 4.3 that training surrogate models in a federated manner can lead to even higher transferability of adversarial examples. However, in the experiment in Section 4.3, we partition the collected data from malicious clients as the federated model and train the source model in a federated manner. In this section, we further evaluate whether this partition information is crucial to the higher transferability. That is, whether different partitions used by the source model affect the transferability of its adversarial examples against the target model. To simulate this setting, we first randomly generate two different partitions with

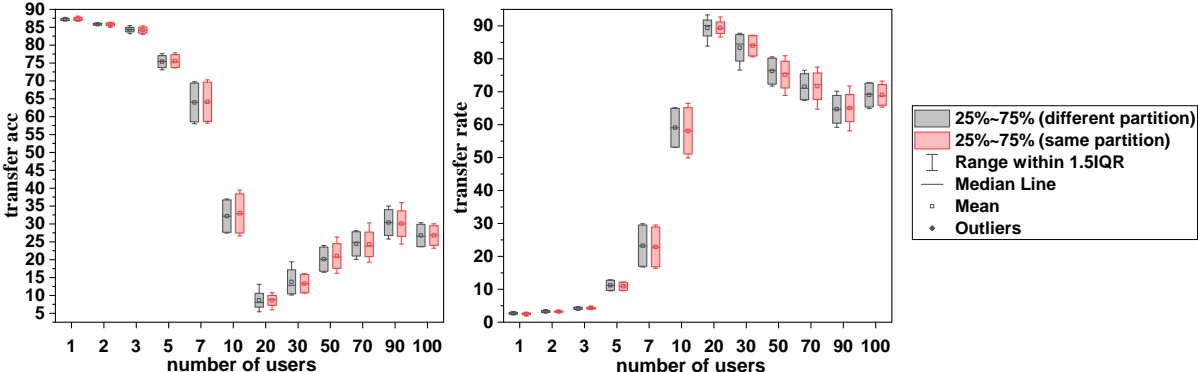

Figure 7: Difference between same partition and different partition; Left: transfer accuracy v.s. number of users' data leveraged in source model training; Right: transfer rate v.s. number of users' data leveraged in source model training

distinct random seeds and then perform the source model training and the target model training on the two different partitions. Then transfer attack is performed with the source model against the target model following the above configuration. We repeat the experiment 4 times with different random seeds and report the averaged results in Figure 7. We observe no significant difference between the same partition and the different partition settings. To further validate this observation, we perform a Hypothesis Test on the obtained results with the Paired Sample T-Test and achieve a p-value of .393 meaning that there is no significant difference. This further demonstrates the possibility of attacking a federated learning system through our proposed attacks and illustrates a higher security risk.

# B  Averaging Leads to Adversarial Robustness with Different Aggregation Methods

We show in Section 5.2 that the averaging operation contributes to the robustness of the FL model and more clients to average per round leads to higher transfer robustness. We further validate that this observation generalizes to different aggregation methods, *i.e.* Krum Blanchard et al. (2017), Geometric Mean Yin et al. (2018) and Trimmed Mean Yin et al. (2018). As per Figure 8, we can see that with all three aggregation methods, there are decreasing trends as the number of averaged users per round increases.

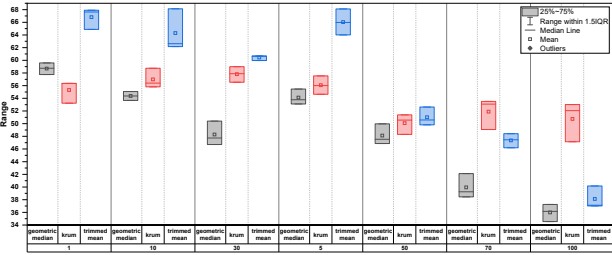

Figure 8: Transfer rate v.s. #clients selected to average in each round with difference aggregation methods.

# C  Statistical Hypothesis Testing on Spearman correlation coefficient

In Section 5.1, we elucidate the correlation between the degree of decentralization and the heterogeneity of training data with the transferability of adversarial examples. The findings denote that, with more decentralized, heterogeneous data, the federated model is more robust to transfer attack. Furthermore, Section 5.2 portrays a discernible inverse relationship between the number of clients and the average transfer success rate, as depicted through box plots, which illustrate the averaging operation leads to better robustness against adversarial examples.

To statistically validate these correlations, we perform two-tailed Hypothesis Testing on Spearman correlation coefficient. To conduct Hypothesis Testing for Spearman correlation coefficient on a specified correlation, we first calculate the Spearman correlation coefficient $\rho$ on the two sets of points (*e.g.*, T.Rate and Dirichlet $\alpha$):

$$\rho = \frac{cov(X,Y)}{\sigma(X)\sigma(Y)}$$

where $cov(\cdot,\cdot)$ denotes the covariance and $\sigma(\cdot)$ represents the standard deviation. To perform the Hypothesis Test, we first have the Null Hypothesis $H_0$ and Alternate Hypothesis $H_a$:

$$H_0 : \rho = 0$$
$$H_a : \rho \neq 0$$

We choose the significance level to be 0.1, which means we reject $H_0$ is p-value is smaller than 0.1. We report the p-value and Spearman correlation coefficient in Table 3.

We can see, as reported in Section 5.1 and 5.2, all experiments except the CNN experiments on dirichlet $\alpha$ and unbalance sgm can deomenstrates significant correlation under a significance level of 0.1. More to the point, we report numerous correlations with p-value less than .001 (significant under level of .001). This Hypothesis Testing validated the findings of our investigation.

Table 3: Spearman correlation coefficient and p-value

| Architecture | X | p-value (two-tailed) | Spearman coefficient |
|---|---|---|---|
| | dirichlet $\alpha$ | .006 | .59 |
| | unbalance sgm | .05 | -.44 |
| | number of user in partition | .003 | -.63 |
| ResNet50 | maximum number of classes (10 users) | < .001 | 0.80 |
| | maximum number of classes (100 users) | < .001 | .76 |
| | number of user to average (30 users) | < .001 | -.71 |
| | number of user to average (centralized) | < .001 | -.79 |
| | dirichlet $\alpha$ | .76 | .074 |
| | unbalance sgm | .84 | .049 |
| | number of user in partition | < .001 | -.83 |
| CNN | maximum number of classes (10 users) | .009 | .45 |
| | maximum number of classes (100 users) | .005 | .67 |
| | number of user to average (30 users) | < .001 | -.77 |
| | number of user to average (centralized) | < .001 | -.83 |

# D  Proof of Theorem 6.1

Following assumptions from Li et al. (2020b) and additional assumptions for adversarial transferability, we provide a low bound for $R$.

**Assumption 1**  $\partial f(x;\theta)/\partial x = \theta^T \rho(x,\theta)$ where $\rho(x,\theta)$ is an arbitrary function in the forms of $\rho(x,\theta) :$ $\mathbb{R}^{1 \times p} \times \mathbb{R}^{p \times 1} \to \mathbb{R}$.

There are many functions following our standards (*e.g.*, $\ell_2$-norm regularized linear regression, logistic regression and softmax classifier). For the rest of our discussion, we define $\nabla l(f(x;\theta),y) = \partial f(x;\theta)/\partial x$

**Lemma D.1.** *With Assumption 1, we have*

$$R(x,y,\theta,\theta') = \frac{\theta^T \theta'}{\|\theta\|_2 \|\theta'\|_2} \leq 1 \tag{7}$$

With Lemma D.1, $\theta$ can be directly measured by the cosine similarity of the parameters $\theta$ and $\theta'$. Furthermore, as we need to offer a discussion regarding multiple aspects of the model, such as the loss, the parameters, and the data, we follow the previous convention Li et al. (2020b) to focus on a narrower scope the model family:

**Assumption 2** Model $f$ is in the form of $f(\theta) = \sum_{x \in X}(x\theta)^2$

Notice that this is not a significant deviation from previous studies Li et al. (2020b) that focus on $f(\theta) = \theta^T A\theta$ for detailed investigation of the parameter behaviors.

**Assumption 3** The covariance of the samples we study is positive semidefinite, i.e., $X^T X \succcurlyeq \mathbf{I}$

**Assumption 4:** The loss function $l$ is L-smooth: for all $v$ and $w$, $l(v) \leq l(w) + (v-w)^T \nabla l(w) + \frac{L}{2}\|v-w\|_2^2$.

**Assumption 5:** The loss function $l$ is $\mu$-strongly convex: for all $v$ and $w$, $l(v) \geq l(w) + (v-w)^T \nabla l(w) + \frac{\mu}{2}\|v-w\|_2^2$.

**Assumption 6:** Let $\xi_t^k$ be sampled from the $k$-th device's local data uniformly at random in iteration $t$. The variance of stochastic gradients in each device is bounded by $\sigma_k^2$: $\mathbb{E}\|\nabla l(\xi_t^k;\theta_t^k) - \nabla l(\theta_t^k)\| \leq \sigma_k^2$

**Assumption 7:** The expected squared norm of stochastic gradients is uniformly bounded, *i.e.*, $\mathbb{E}\|\nabla l(\xi_t^k;\theta_t^k)\|^2 \leq G^2$ for all $k = 1, \cdots, N$ and $t = 1, \cdots, T-1$.

**Assumption 8:** We assume the federated algorithm is FedAvg. Assume $S_t$ contains a subset of $K$ indices randomly selected with replacement according to the sampling probabilities $p_1, \cdots, p_N$. The aggregation step of FedAvg performs $\theta_t \leftarrow \frac{1}{K}\sum_{k \in S_t} \theta_t^k$. $\theta_t^k$ denotes the parameters of device $k$ at iteration $t$.

We use Theorem 2 from Li et al. (2020b) as our lemma to prove our Theorem 6.1.

**Lemma D.2.** *With assumption 4 to 8 and $L$, $\mu$, $\sigma_k$ and $G$ be defined therein. Let $L^\star$ denote the minimum loss obtained by optimal estimation from the centralized model and $l(\theta')$ denotes the loss of the federated model. Then*

$$\mathbb{E}\big[l(\theta')\big] - L^\star \leq \frac{\kappa}{\gamma + T - 1}\Big(\frac{2(B+C)}{\mu} + \frac{\mu\gamma}{2}\mathbb{E}\|\theta_1 - \theta_\star\|\Big) \tag{8}$$

*where $L, \mu, \sigma_k, G$ is defined in the assumptions, $\kappa = \frac{L}{\mu}$, $\gamma = \max\{8\kappa, E\}$, $B = \sum_{k=1}^{N} p_k^2 \sigma_k^2 + 6L\Gamma + 8(E-1)^2 G^2$ and $C = \frac{4}{K}E^2 G^2$. $\theta_1$ is the parameter after one step update of SGD. $\theta_\star$ is the optimal parameter for centralized model.*

Now, we prove Theorem 6.1.

*Proof.* We first write out

$$l(\theta') - L^\star = \theta'^T X^T X \theta' - \theta_\star^T X^T X \theta_\star \tag{9}$$

$$= (\theta' - \theta_\star)^T X^T X (\theta' - \theta_\star) \tag{10}$$

$$\geq \|\theta'\|_2 \|\theta_\star\|_2 \tag{11}$$

when $X^T X$ is p.s.d.

On the other hand, we can write $l(\theta') = L^\star + \epsilon$ where $\epsilon > 0$. Due to the construction of our model, we can write

$$\theta' = (X^T X)^{-1} X^T \sqrt{(X\theta_\star)^2 + \epsilon} \tag{12}$$

by solving a linear system. Thus, we can get

$$\theta_\star^T \theta' = \theta_\star^T (X^T X)^{-1} X^T \sqrt{(X\theta_\star)^2 + \epsilon} \tag{13}$$

$$\geq \theta_\star^T (X^T X)^{-1} X^T X \theta_\star \tag{14}$$

$$= \theta_\star^T \theta_\star \tag{15}$$

Thus, by connecting the above terms, we will have

$$R(x, y, \theta_\star, \theta') = \frac{\theta'^T \theta_\star}{\|\theta'\|_2 \|\theta_\star\|_2} \geq \frac{\theta_\star^T \theta_\star}{l(\theta') - L^\star} \tag{16}$$

Finally, by substituting Equation 8 to the denominator, we have

$$\mathbb{E}[R(x, y, \theta', \theta_\star)] \geq \frac{(\gamma + T - 1)\theta_\star^T \theta_\star}{\kappa \left( \dfrac{2(B + C)}{\mu} + \dfrac{\mu\gamma}{2} \mathbb{E}\|\theta_1 - \theta_\star\|^2 \right)} \tag{17}$$

$$= \frac{2\mu(\gamma + T - 1)\theta_\star^T \theta_\star}{4(B + C)\kappa + \mu^2 \gamma \kappa \mathbb{E}\|\theta_1 - \theta_\star\|^2} \tag{18}$$

$$\square$$

# E   Proof of Theorem 6.5

*Proof.* We adopt the Bias-Variance decomposition Geman et al. (1992) to provide theoretical justification for the observation in Section 5.2. Let $\mathcal{T}$ denote the train set $\mathcal{T} = (X, Y)$ sampled from some data distribution and $x'$ denote the adversarial example following some adversarial distribution. First, we give the Bias-Variance decomposition as follows:

$$\mathbb{E}_{x',y} \mathbb{E}_{\mathcal{T}}[\|y - f(x'; \mathcal{T})\|_2^2] = \underbrace{\mathbb{E}_{x',y}[\|y - \bar{f}(x')\|_2^2]}_{\text{Bias}^2} \tag{19}$$

$$+ \underbrace{\mathbb{E}_{x', \mathcal{T}}[\|\bar{f}(x') - f(x'; \mathcal{T})\|_2^2]}_{\text{Variance}}$$

where $f(\cdot; \mathcal{T})$ denotes a model $f$ trained on dataset $\mathcal{T}$. $\bar{f}(\cdot) = \mathbb{E}_{\mathcal{T}}[f(\cdot; \mathcal{T})]$ is the expected the model over the data distribution of $\mathcal{T}$. By using an average of multiple model $f(\cdot; \mathcal{T})$, denoted by $f_n = \frac{1}{n} \sum_{i=1}^n f_i(\cdot; \mathcal{T}_i)$ and with $\bar{f}_n(\cdot) = \mathbb{E}_{\mathcal{T}}[\frac{1}{n} \sum_{i=1}^n f_i(\cdot; \mathcal{T})] = \bar{f}(\cdot)$, we can show the following:

$$\mathbb{E}_{x',y} \mathbb{E}_{\mathcal{T}}[\|y - f_n(x')\|_2^2] = \underbrace{\mathbb{E}_{x',y}[\|y - \bar{f}_n(x')\|_2^2]}_{\text{Bias}^2} \tag{20}$$

$$+ \underbrace{\mathbb{E}_{x', \mathcal{T}}[\|\bar{f}_n(x') - f_n(x')\|_2^2]}_{\text{Variance}}$$

$$= \mathbb{E}_{x',y}[\|y - \bar{f}(x')\|_2^2] + \frac{1}{n} \mathbb{E}_{x', \mathcal{T}}[\|\bar{f}(x') - f(x', \mathcal{T})\|_2^2]$$

Inequality 7 holds due to that $\text{Var}(\frac{1}{n} \sum_{i=1}^n f_i(\cdot; \mathcal{T}_i)) = \frac{1}{n^2} \text{Var}(\sum_{i=1}^n f_i(\cdot; \mathcal{T}_i))$ and since each $f_i$ is trained independently, $\frac{1}{n^2} \text{Var}(\sum_{i=1}^n f_i(\cdot; \mathcal{T}_i)) = \frac{1}{n^2} \sum_{i=1}^n \text{Var}(f(\cdot; \mathcal{T})) = \frac{1}{n} \text{Var}(f(\cdot; \mathcal{T}))$. $\square$

# F   Attacking Communication-efficient FL Methods

We also explore whether our attack performs well on communication-efficient federated methods such as DAdaQuant Hönig et al. (2022) and FedPAQ Reisizadeh et al. (2020b). Specifically, we compare the standard FedAvg, Krum, Trimmed Mean and FedPAQ Reisizadeh et al. (2020b).

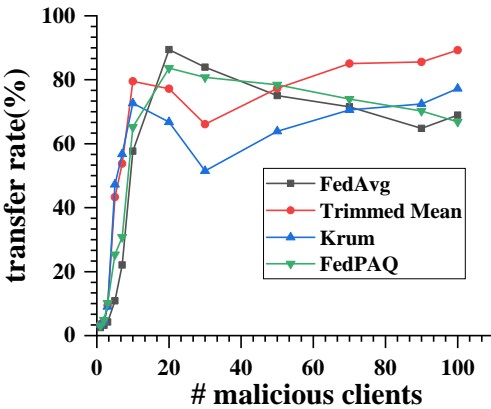

Figure 9: Attacking communication-efficient FL.

## G  Comparison between transfer attack and attack using traces of training.

In this section, we explore using the traces of training (checkpoints during the training of the federated model) to attack and compare with the transfer attack presented in the paper. We must emphasize that the former is a different threat model with the capability of keeping traces of the model at different stages of training. That is to say, the attack is a purely white-box one.

We present the attack result using checkpoints of different stages of the training as shown in Tab 4. We early stopped the model when the accuracy reaches 90% which is around 75 epochs. Then we use the checkpoints from 20, 40 and 60 epochs to perform the attack.

|  | white-box | 20e | 40e | 60e |
| --- | --- | --- | --- | --- |
| transfer rate | 91.48 | 89.98 | 91.45 | 94.66 |
|  | 1% transfer | 10% transfer | 30% transfer | 100% transfer |
| transfer rate | 2.49 | 57.69 | 83.91 | 68.94 |

Table 4: Comparison between using checkpoint (traces of training) attack and transfer attack.

## H  linear regression to visualize the correlation

To better demonstrates the correlation between various factors and adversarial transferability, we perform linear regression with hypothesis testing on the experiment results. We plot scatter graph and linear regression line on each of the correlation and corresponding experiment result as shown in the following figures:

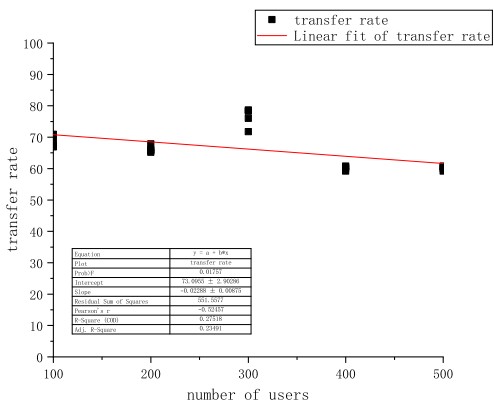

(a) ResNet50: transfer rate v.s. number of total users in the partition.

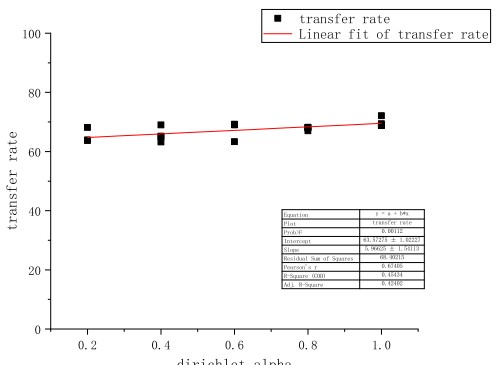

(b) ResNet50: transfer rate v.s. dirichlet alpha.

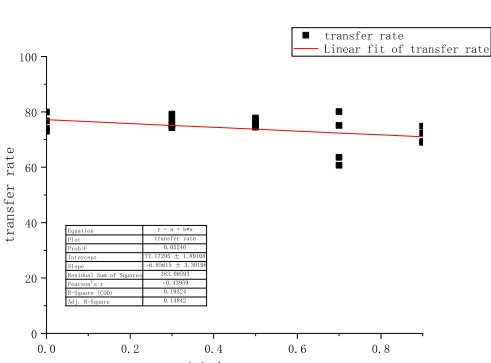

(c) ResNet50: transfer rate v.s. unbalance sgm.

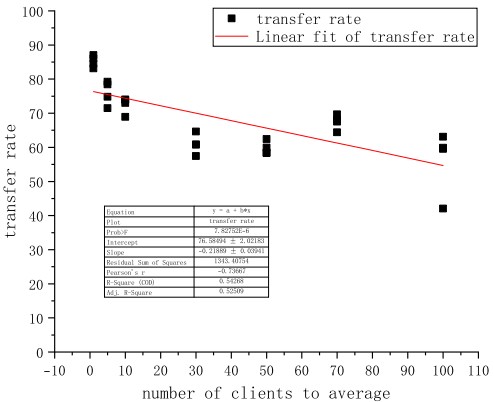

(d) ResNet50: transfer rate v.s. number of users to average per round (source model trained in centralized manner with full training dataset).

Figure 10: Linear regression visualization

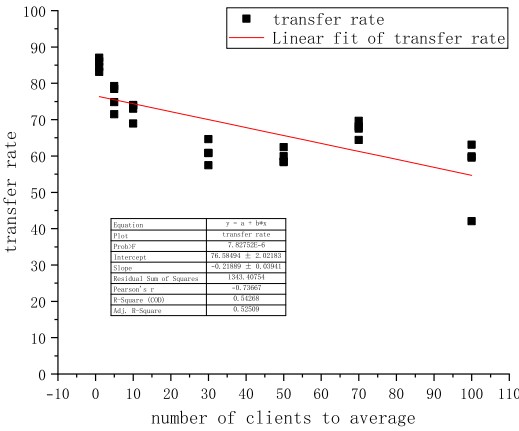

Figure 11: ResNet50: transfer rate v.s. number of users to average per round (source model trained in centralized manner with 30 client's data).

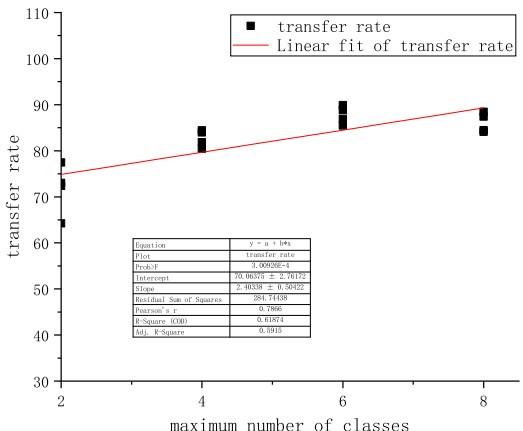

Figure 12: ResNet50: transfer rate v.s. maximum number of classes per user (10 users).

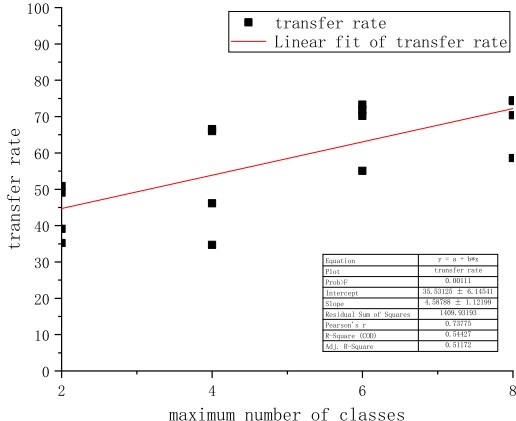

Figure 13: ResNet50: transfer rate v.s. the maximum number of classes per user (100 users).

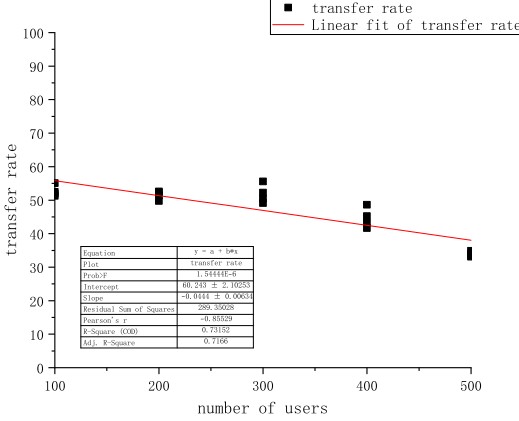

Figure 14: CNN: transfer rate v.s. the number of total users in the partition.

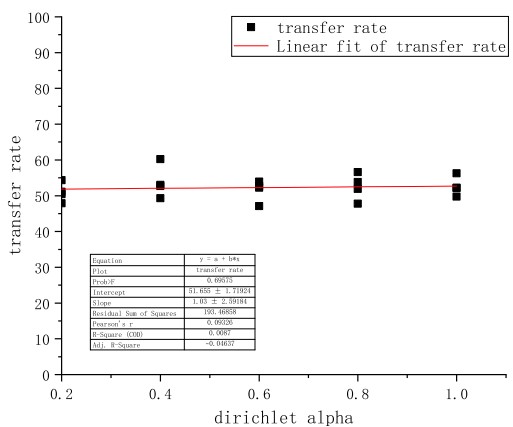

Figure 15: CNN: transfer rate v.s. dirichlet alpha.

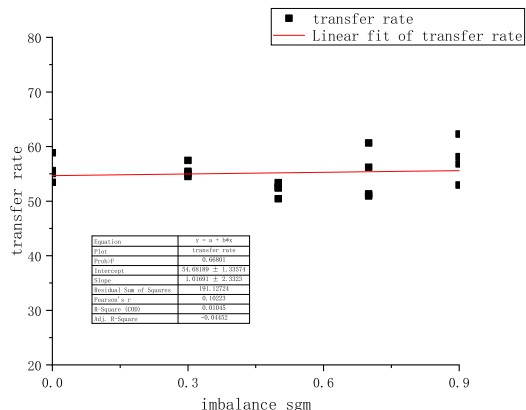

Figure 16: CNN: transfer rate v.s. unbalance sgm.

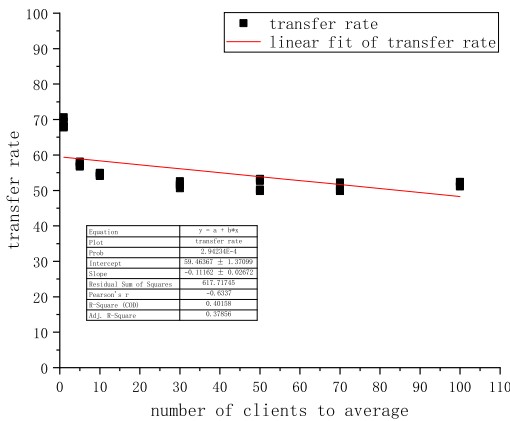

Figure 17: CNN: transfer rate v.s. number of users to average per round (source model trained in centralized manner with full training dataset).

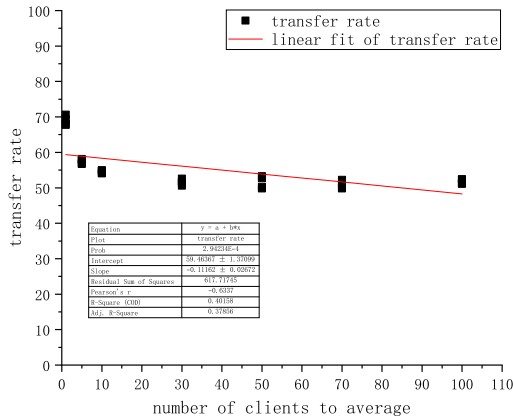

Figure 18: CNN: transfer rate v.s. number of users to average per round (source model trained in centralized manner with 30 client's data).

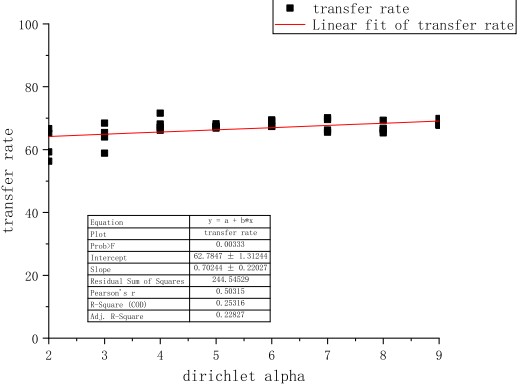

Figure 19: CNN: transfer rate v.s. maximum number of classes per user (10 users).

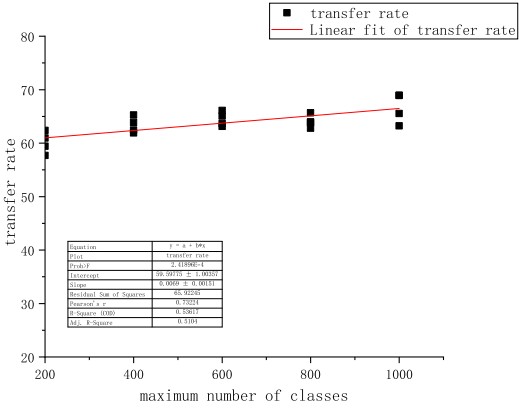

Figure 20: CNN: transfer rate v.s. the maximum number of classes per user (100 users).

