# OpenReview forum: "Towards Understanding Adversarial Transferability in Federated Learning"
_TMLR — Accepted by TMLR_

### Review · Reviewer_jQmf · 2024-09-14

**Summary Of Contributions:**

This work considers inference/validation-time attacks where adversarial perturbations are constructed from a fraction of the training data. The authors operate in a black-box setting: they do not assume the adversary has knowledge of the target's training procedure. Conversely, the authors assume the adversary is passive during training; it is only during afterward that they act by perturbing a validation set. Success of the attack is measured by a transferability rate (T.Rate) metric and success of the target model is measured by transfer accuracy (T.Acc) metric. The authors' main interest is in the success or failure of these attacks against federated learning.

One baseline finding is that white-box attacks against centralized models are more successful than those against federated models with the same accuracy. A second baseline finding is that adversarial examples generated by federated training are highly transferable in the unlimited data setting; as shown in Fig 1b and 1c, it holds true in the authors' limited-data setting

More generally, the authors find (in Section 4.3) that only a small fraction of corrupted clients are enough to attack federated models. As an example, they show T.acc drops to (near) 0 with ~30/100 users on CIFAR10 (Figure 1a)

On the other hand, introducing more "federated-ness" in the training seems to mitigate transferability. Specifically, the authors give experimental evidence that T.Rate is reduced by dividing the dataset across more partitions and by making the distributions of data more heterogeneous.

**Audience:**

Yes

**Claims And Evidence:**

Yes

**Requested Changes:**

**Post-rebuttal comment**: the authors responded to my questions and requests to my satisfaction

---

*Clarifications*

Is the following a correct description of the interactions between the participants?
1. Adversary corrupts some fraction of clients/devices
2. All clients/devices receive data and honestly participate in the training process of M
3. Meanwhile, adversary feeds the data of the corrupted clients/devices into its own (ML training) algorithm to obtain some perturbation $\delta$
4. Adversary receives a set of examples and corrupts them using $\delta$

If it is a correct description, consider including a summary like it to make it easier for the reader to understand all the important features of the setting at a glance. As it is, this information is scattered throughout the paper.

Please clarify the threat model in Section 4.2. In particular, how does it differ from the steps above?

Please clarify whether the data of the corrupted clients is chosen by the adversary or if they come from the same distribution as the others. Similarly, how is the validation set created?

---

*Questions*

Is the boldface conclusion in Section 4.3 actually unique to FL target models? That is, if we were to repeat the experiments with centralized training, would we get reduced transfer rate?

The experiments were on CIFAR10. Do you conjecture that similar behavior would occur on different datasets? Why or why not? And what about different types of classifiers (e.g. different architectures)?

---

*Presentation*: Consider re-distributing the information conveyed by the figures. For example Figure 1c has information relevant to "Training substitute model in a federated manner" "Boosting Transfer Rate" and "Advanced transfer attack" which is a bit much for one subfigure.

Do "source model" "substitute model" and "surrogate model" all refer to the same thing?

---
*Smaller editing errors*:
- In equation (4) where does the data of the malicious users play a role? It seems x is some fixed value.
- typo: equation (4) should read argmax instead of argmin
- typo: Threat Model paragraph begins with the word "follwing"
- typo: Figure 1 (b)'s caption says "ResNet50" when the title says "CNN"

**Strengths And Weaknesses:**

Strengths
- The threat model appears to be a natural one.
- The related work section is fairly rich and I appreciate the flagging of Key Differences.
- I appreciate the variety of experiments

Weaknesses [see also Requested Changes]
- Discussion and Conclusion sections are quite limited
- Clarity of exposition can be improved in multiple places in the submission.

---

> ### Author Response · Authors · 2024-10-09
> **Response to reviewer jQmf**
>
> ```>>> Q1``` Is the following a correct description of the interactions between the participants?
> - Adversary corrupts some fraction of clients/devices
> - All clients/devices receive data and honestly participate in the training process of M
> - Meanwhile, adversary feeds the data of the corrupted clients/devices into its own (ML training) algorithm to obtain some perturbation $\delta$
> - Adversary receives a set of examples and corrupts them using $\delta$
>
> ```>>> A1``` Thanks for the question. Yes, this is a correct description of the attack setting. We will provide a algorithm flow in the revised version of the paper for better comprehension.
>
> ---
>
> ```>>> Q2``` Please clarify the threat model in Section 4.2. In particular, how does it differ from the steps above?
>
> ```>>> A2``` Thanks for your question. The only difference is the fraction of clients corrupted by the attacker.
>
> - In Section 4.2, we perform a black-box transfer attack, where all the training data is used to train a centralized surrogate model.
> - In our attack setting, attackers only possess a fraction of the users participating in the FL training.
>
> ---
>
> ```>>> Q3``` Please clarify whether the data of the corrupted clients is chosen by the adversary or if they come from the same distribution as the others.
>
> ```>>> A3``` For most of the experiments, the corrupted clients use data from the same distribution as the others. By utilizing data from the same distribution as the benign clients, we essentially eliminate the possibility of being detected, as there are no observable differences between malicious and benign clients. To further evaluate our attack setting, we also conduct experiments to explore non-IID cases where the malicious clients' data differ from those of the benign clients. As shown in Section 5 and Figure 3, this leads to lower attack performance.
>
> ---
>
> ```>>> Q4``` Similarly, how is the validation set created?
>
> ```>>> A4``` We use the default test set of CIFAR10, CIFAR100 and SVHN dataset as the validation set.
>
> ---
>
> ```>>> Q5``` Is the boldface conclusion in Section 4.3 actually unique to FL target models? That is, if we were to repeat the experiments with centralized training, would we get a reduced transfer rate?
>
> ```>>> A5``` We present a comparison in the following figure https://hackmd.io/_uploads/HkQYdjbkkg.jpg.
> The boldface conclusion in Section 4.3 applies to the centralized target model as well, with an even higher attack success rate. This is because the attacker's surrogate model is trained in the same centralized manner. We also show the result with the surrogate model trained in an FL manner for comparison.
>
> ---
>
> ```>>> Q6``` The experiments were on CIFAR10. Do you conjecture that similar behavior would occur on different datasets? Why or why not? And what about different types of classifiers (e.g. different architectures)?
> ```>>> A6``` Thanks for the concern. Yes, we have included results on CIFAR100, SVHN, and ImageNet-200 (as requested by reviewer i4Nu), which all lead to a similar trend and result.
>
> As for different types of classifiers and architectures, we have also provided results with CNN and VGG16 (see CNN part of Table 1, 2, (a) and (b) of Figure 1, 3, 4, \(c) and (d) of Figure 6) and Auto-attack, skip-attack (different adversarial attacks, see \(c) of Figure 1).
>
> ---
>
> ```>>> Q7``` In equation (4) where does the data of the malicious users play a role? It seems x is some fixed value.
>
> ```>>> A7``` Equation (4) is the optimization to get the perturbation $\delta$, which is a maximization of loss with respect to the $\delta$ with some fixed $x$. Essentially, it means that, for some input $x$, we want to compute a perturbation $\delta$ that gives the largest loss with some constraint $d(x + \delta, x) < \epsilon$.
>
> ---
>
> ```>>> Q8``` Presentation: Consider re-distributing the information conveyed by the figures. For example Figure 1c has information relevant to "Training substitute model in a federated manner" "Boosting Transfer Rate" and "Advanced transfer attack" which is a bit much for one subfigure.
> ```>>> A8``` Thank you for the constructive comments! We will try our best to re-distribute the contents for better comprehension in the final version of the paper.
>
> ---
>
> ```>>> Q9``` Do "source model" "substitute model" and "surrogate model" all refer to the same thing?
> ```>>> A9``` Thanks for raising this question. Yes, "source model" "substitute model" and "surrogate model" all refer to the model trained by the attacker to launch the transfer attack. We have revised the "substitute model" to the "surrogate model". We use the "source model" and "surrogate model" following terminology from prior work on black-box adversarial attacks.
>
> ---
>
> ```>>> Q10``` Typos.
> ```>>> A10``` Thank you for pointing them out. All typos are fixed in the revised version.

---

### Review · Reviewer_i4Nu · 2024-09-16

**Summary Of Contributions:**

This paper analyze the "transfer-based black-box attacks", where a subset of training data is assumed to be used to train an adversarial attack model. The main contribution includes:
1. Empirically observed that federated model is more robust under white-box attack than centralised models
2. Empirically observed that models trained with the subset of train set can be used for adversarial attack
3. Empirically investigate the methods that is able to increase/decrease model's robustness to transfer-based black-box attacks
4. Provides some theoretical analyze on the transfer-based black-box attacks on federated model/centralised model

**Audience:**

No

**Claims And Evidence:**

Yes

**Requested Changes:**

The contribution looks good empirically, but I think this paper may not of interest for machine learning community.

**Strengths And Weaknesses:**

Pros:
1. Figures looks good
2. Paper well written

Cons:
1. Not very related to machine learning: most of the contributions are empirical observations and analysis, the contents are not very related to machine learning
2. Insufficient experiments: experiments mostly on small scale datasets (CIFAR10/100, SVHN). The Federate learning has it's most potential usage in large scale datasets. Results (especially empirical observations) on small scale dataset is not convincing enough.

---

> ### Comment · Reviewer_i4Nu · 2024-10-01
> **About the reseach venue**
>
> Just to let the authors know, I recognize that maybe I'm a bit too harsh on the relevance of the topic to machine learning. So that's not my concern anymore.
>
> If the author could address the concerns of the other reviewers or possibly provide experimental results on a bit larger scale datasets like ImageNet-200, I'm happy to recommend acceptance.

---

> ### Author Response · Authors · 2024-10-11
> **Response to reviewer i4Nu**
>
> Thank you for the comment. We have conducted additional experiments on ImageNet200 to validate all the results.
>
> First, we reassessed the robustness at equivalent accuracy levels on ImageNet200. As demonstrated in Section 4.1, FL exhibits superior robustness compared to its centralized counterpart when achieving similar clean accuracy. Our new experiments confirm that this conclusion holds true. For these evaluations, we used a setup similar to that in the paper's PGD attack, with an adjusted epsilon of 2/255.
>
> |  Paradigm   | Acc | Adv acc |
> | -------- | ------- |   ------- |
> | Federated |     55.03      |    13.42    |
> | Centralized |     55.05       |   3.68     |
>
> Using adversarial training (FGSM with ε = 8/255), we observe results consistent with those presented in our manuscript: when the clean accuracy is comparable, federated learning demonstrates greater robustness against adversarial attacks. In both the federated and centralized settings, early stopping was applied once the models achiee 50% clean accuracy.
>
> |  Paradigm   | Acc | Adv acc |
> | -------- | ------- |   ------- |
> | Federated |    50.18       |  38.31    |
> | Centralized |   50.04      |   31.20    |
>
> Next, we conducted transfer attacks following the methodology outlined in Section 4.2 and Table 2 of the manuscript. The results for ImageNet200 are presented in the table below (left: transfer accuracy, right: transfer rate).
>
> |     | federated | centralized |
> | -------- | ------- |   ------- |
> | **federated** |    2.29 / 95.60      |  6.52 / 86.74 |
> | **centralized** |     22.72 / 54.00   |   8.27 / 83.13   |
>
> We can conclude exactly the same result:
> 1. Adversarial examples generated by the federated model are highly transferable to both the federated and centralized model
> 2. Adversarial examples generated by the centralized model exhibit less transferability.
> 3. The transfer rate of adversarial examples between models trained under the same paradigms is larger than models trained under different paradigms.
>
> We refer to Section 4.2 for more discussion.
>
> We conducted additional experiments on the ImageNet200 dataset to validate our attack setup and assess the two intrinsic properties that lead to robustness. The results are shown in the following figure https://hackmd.io/_uploads/SyE-_8Lyyx.jpg.
>
> In Figure (a), we illustrate our attack using a limited data setup on the ImageNet200 dataset. With 10% of clients compromised, we achieve a transfer rate of over 30%, which increases to 45% when 20% of clients are compromised. This underscores the significant risk our attack setup poses to federated learning systems.
>
> Figure (b) demonstrates that averaging more clients per round improves transfer robustness in FL models, as the transfer rate decreases with an increasing number of clients.
>
> Figures (c) and (d) further validate that both decentralization and heterogeneity contribute to enhanced transfer robustness.
>
> Overall, the results on the large-scale ImageNet200 dataset consistently support our conclusions, demonstrating that our findings generalize across datasets, architectures, and FL methods.

---

### Review · Reviewer_g9J4 · 2024-09-17

**Summary Of Contributions:**

This paper analyzes the impact of black-box transfer attacks in a federated learning scenario when a set of benign clients turn malicious after the original FL training. The results demonstrate that as compared to a centralized setting,FL systems are more robust to such adversarial attacks. The authors attribute this to heterogeneous data distributions over the participating clients and the averaging operation common in FL.

**Audience:**

Yes

**Claims And Evidence:**

No

**Requested Changes:**

1. Since the malicious clients could participate in FL training by pretending to be benign, would the transfer attack accuracy increase if the malicious clients kept traces of the model at different stages of training and utilized that to launch an attack? Please clarify and provide a discussion on this scenario.

2. In Figure 1(d), it is unclear why the attack success decreases over training, please provide a discussion on the same.

3. In Figure 3(b) on the trends with Dirichlet coefficient alpha, the trend that the transferability increases with decreasing degree of heterogeneity does not hold.

4. The robustness of the averaging operation is interesting, but it is equally important to evaluate the impact of sampling the benign clients that turn malicious. In other words, can you provide some experiments to discuss the impact of different rates of benign client sampling during FL training upon the transfer rate of the attack?

5. Communication efficiency is another crucial aspect of federated learning, with several communication compression techniques proposed to reduce the communication overhead [1]. Can you comment upon the attack transfer rate assuming the FL training was done in a communication-efficient manner?

[1] R. Honig et al., "DAdaQuant: Doubly-adaptive quantization for communication-efficient Federated Learning", ICML'22.

**Strengths And Weaknesses:**

Strengths:

1. This work studies a unique FL security risk and conducts extensive experiments to judge the degree of robustness inherent to such systems against adversarial transfer attacks.

Weaknesses:

1. I am concerned about the motivation and broader applicability for this work as mentioned on page 2: "Stemming from the above scenarios, we propose a simple yet practical assumption: the attacker possesses a limited amount of the users’ data but no knowledge about the target model or the full training set". If the attacker participated as a benign client in the FL training and then turned malicious, it would end up having access to the target model as communicated by the server. In that scenario, I am wondering if model inversion attacks seem more plausible. Gboard example makes sense in a setting where models are protected by a layer of encryption as mentioned, but my concerns remain about the broad applicability of the observations presented in this paper.

2. Another assumption I have concerns with is the ability of several malicious clients to jointly train a model in a federated manner to increase the transfer rate of the attack. How do these malicious clients access the server after the original FL training to launch an attack while the server cannot detect their intentions? In my understanding, in FL the server holds control over what kind of training should be done and when, hence I am not convinced that a set of malicious clients can decide to use PGD without the server's awareness. Please provide some clarifications in case I misunderstood the setting described on page 6 under "Training substitute model in a federated manner".

3. Some experimental results and claims are unclear (see Requested Changes).

---

> ### Author Response · Authors · 2024-10-09
> **Response to reviewer g9J4**
>
> We thank the reviewer for the valuable comments and suggestions.
>
> ```>>> Q1``` I am concerned about the motivation and broader applicability of this work as mentioned on page 2: "Stemming from the above scenarios, we propose a simple yet practical assumption: the attacker possesses a limited amount of the users’ data but no knowledge about the target model or the full training set". If the attacker participated as a benign client in the FL training and then turned malicious, it would end up having access to the target model as communicated by the server. In that scenario, I am wondering if model inversion attacks seem more plausible. Gboard example makes sense in a setting where models are protected by a layer of encryption as mentioned, but my concerns remain about the broad applicability of the observations presented in this paper.
>
> ```>>> A1``` Thank you for raising your concern. While we acknoledge that the common assumption behind Federated Learning is that malicious users will gain access to the target model as communicated by the server. However, we believe this is not usually the practice case, especially when it's a To-Customer service. Examples such as Gboard [1, 2, 3], Nvidia Clara [4], Siri [5] and applications of FL in Safari [5] demonstrate that on top of the FL, there will be additional layers (or even encryption layers) that prevent the attacker from accessing the model weights easily. For instance, as mentioned in [3], multiple layers of encryption (e.g. cryptographic secure multi-party computation (MPC) protocol) have been applied. In our attack setting, however, there is no need to access the well-protected model weights, and the attack also avoids being detected by any defense algorithms, as it operates as benign.
>
> Thus, while model inversion attacks may be plausible in certain scenarios, our work focuses on a broader context where such protections are in place, and yet vulnerabilities remain exploitable.
>
> [1] Xu, Zheng, et al. "Federated learning of gboard language models with differential privacy." arXiv preprint arXiv:2305.18465 (2023).
> [2] Yang, Timothy, et al. "Applied federated learning: Improving google keyboard query suggestions." arXiv preprint arXiv:1812.02903 (2018).
> [3] Zhang, Yuanbo, et al. "Private federated learning in gboard." arXiv preprint arXiv:2306.14793 (2023).
> [4] NVIDIA Clara. NVIDIA. Published 2024. Accessed October 7, 2024. https://www.nvidia.com/en-us/clara/.
> [5] Learning with Privacy at Scale Differential Privacy Team, Apple. https://docs-assets.developer.apple.com/ml-research/papers/learning-with-privacy-at-scale.pdf.
> ‌
>
> ---
>
> ```>>> Q2``` Another assumption I have concerns with is the ability of several malicious clients to jointly train a model in a federated manner to increase the transfer rate of the attack. How do these malicious clients access the server after the original FL training to launch an attack while the server cannot detect their intentions? In my understanding, in FL the server holds control over what kind of training should be done and when, hence I am not convinced that a set of malicious clients can decide to use PGD without the server's awareness. Please provide some clarifications in case I misunderstood the setting described on page 6 under "Training substitute model in a federated manner".
>
> ```>>> A2``` Thanks for raising this concern. There seems indeed a misunderstanding. The term "training a surrogate model in a federated manner" refers to malicious clients simulating the federated learning (FL) process within their group using data under their control. For example, if there are 10 malicious users among 100 total users, these attackers will train a surrogate model using FedAvg on the combined data of the 10 malicious users: simulating FL among themselves. This will lead to a 10-user FL training, where each user controls the exact data of this malicious user used to train the target model. This means that the adversary simulates this FL behavior on its own, as we observe that FL training leads to better surrogate model to attack the FL target model.

---

> ### Author Response · Authors · 2024-10-09
> **Response to reviewer g9J4**
>
> ```>>> Q3``` Since the malicious clients could participate in FL training by pretending to be benign, would the transfer attack accuracy increase if the malicious clients kept traces of the model at different stages of training and utilized that to launch an attack? Please clarify and provide a discussion on this scenario.
>
> ```>>> A3``` We present the attack result using checkpoints of different stages of the training as shown below. We early stopped the model when the accuracy reached 90% which is around 75 epochs. Then we use the checkpoints from 20, 40 and 60 epochs to attack.
>
> |     | white-box | 20e | 40e | 60e |
> | -------- | ------- |  ------- |  ------- |   ------- |
> | transfer rate  | 91.48 | 89.98 | 91.45 | 94.66 |  |  |
> | | **1% transfer** | **10% transfer** | **30% transfer** | **100% transfer** |
> | transfer rate | 2.49 | 57.69 | 83.91 | 68.94 |
>
> We must emphasize here that **if the threat model is equiped with the ability to keep traces of the model at different stages of training, the attack setup will be a purely white-box attack.**
>
> Despite the white-box attack achieving better attack performance, we kindly remind that the capability of the threat model is **completely different** than ours, and impractical as discussed in Q1. Our attack, without access to any model weights or even performing malicious acts, achieves 57.69% with 10% malicious clients and this number increases to 83.91% when 30% of the clients are corrupted.
>
> ---
>
> ```>>> Q4``` In Figure 1(d), it is unclear why the attack success decreases over training, please provide a discussion on the same.
>
> ```>>> A4``` Thank you for raising this comment. We kindly remind that we have provided discussions on this phenomenon in Section 4.3 (See "To further explain observation 1 and an intriguing phenomenon that the T.Rate of ResNet50 model rises to the peak and then decreases").
>
> We provide a hypothesis to explain this phenomenon: When the number of clients used to train the source model is small, the clean accuracy of the source model is also low, leading to a large discrepancy in the decision boundary. Increasing the number of users used in the source model minimizes such discrepancy until the amount of data is sucient to train a source model with similar accuracy. At this point, the difference between the federated and centralized becomes the dominant factor aecting the transferability since the source model is trained in the centralized paradigm.
>
> We validate this hypothesis by training the source model in an FL manner. We illustrate that when the source model is trained in FL, the attack success rate keeps increasing as the number of malicious clients increases. This further highlights the security breaches identified by our attack setting.
>
> ---
>
> ```>>> Q5``` In Figure 3(b) on the trends with Dirichlet coefficient alpha, the trend that the transferability increases with decreasing degree of heterogeneity does not hold.
>
> ```>>> A5``` Thanks for this comment, we agree that the trend does not hold for this particular case, as illustrated in Table 3 of Appendix C, which has a high p-value of 0.76. We will also point out this in the main result in the revised version of the paper.
> ‌
>
> ---
>
> ```>>> Q6``` The robustness of the averaging operation is interesting, but it is equally important to evaluate the impact of sampling the benign clients that turn malicious. In other words, can you provide some experiments to discuss the impact of different rates of benign client sampling during FL training upon the transfer rate of the attack?
>
> ```>>> A6``` Thanks for the suggestion. We have conducted experiments with different fractions of malicious clients, as illustrated in https://hackmd.io/_uploads/ryOKviW1yl.jpg. We show that as the number of benign clients that turn malicious increases, the attack performance will correspondingly increase.

---

> ### Author Response · Authors · 2024-10-09
> **Response to reviewer g9J4**
>
> ```>>> Q7``` Communication efficiency is another crucial aspect of federated learning, with several communication compression techniques proposed to reduce the communication overhead [1]. Can you comment upon the attack transfer rate assuming the FL training was done in a communication-efficient manner?
>
> ```>>> A7``` Thank you for your valuable suggestion. However, since the code from [1] is not publicly available, we have opted to use FedPAQ [2] as a substitute, which serves as an important baseline in [1]. We present the results below, along with other aggregation methods for comparison. Please see https://hackmd.io/_uploads/r1LhTwE1ye.jpg.
>
> We observe a similar trend between FedPAQ and FedAvg, demonstrating that our attack is applicable to a broader range of federated algorithms and systems.
>
> [1] R. Honig et al., "DAdaQuant: Doubly-adaptive quantization for communication-efficient Federated Learning", ICML'22.
> [2] Reisizadeh, Amirhossein, et al. "Fedpaq: A communication-efficient federated learning method with periodic averaging and quantization." International conference on artificial intelligence and statistics. PMLR, 2020.

---

### Author Response · Authors · 2024-10-11
**Overview of rebuttal and revision**

Dear Action Editor and Reviewers,

We would like to express our sincere gratitude to all the reviewers for their thoughtful and constructive feedback. We also apologize for the delay in our response. The additional time was necessary due to the complexity of the rebuttal experiments, which required significantly more time than anticipated. Specifically, one of the reviewers recommended re-running all experiments on a substantially larger dataset, ImageNet200, which extended the process.

We have addressed all the comments from the three reviewers and revised the manuscript accordingly. The key revisions are as follows:

1. We clarified that the trend observed in Figure 3(b) is not statistically significant (see the revised Section 5.1, highlighted in red).
1. We have included an algorithm to illustrate the process of our attack (see Algorithm 1).
2. All identified typos have been corrected (with changes highlighted in red).
3. We replaced all occurrences of the "substitute model" with "surrogate model" for improved clarity and consistency.

Additionally, **all the experiments and results from the rebuttal will be incorporated into the final version of the paper.**

If there are any comments or questions, please do not hesitate to bring them to our attention, and we will address them promptly. We welcome any further discussion on any aspects that may require clarification.

---

### Decision · Action_Editor_rF4t · 2024-10-23

**Recommendation:** Accept as is

**Comment:**

The recommendation is based on the reviewers' comments, the action editor's evaluation, and the authors’ response.

This paper studies a practical transfer attack setting in federated learning and discovers improved robustness in such a scenario.
All reviewers find the studied setting novel and the results provide new insights. The authors’ rebuttal has successfully addressed the major concerns of reviewers. Therefore, I recommend acceptance of this submission. I also expect the authors to include the new results and suggested changes during the rebuttal phase to the final version.

**Audience:**

The topic is of broad interest to the audience, as it touches upon federated learning, adversarial robustness, and security.

**Claims And Evidence:**

The empirical evidence justifies the claims on the transfer attack and robustness analysis in federated learning.